# Bayes optimal learning of attention-indexed models

**Fabrizio Boncoraglio**
Statistical Physics of Computation Laboratory
EPFL, Switzerland

**Emanuele Troiani**
Statistical Physics of Computation Laboratory
EPFL, Switzerland

**Vittorio Erba**
Statistical Physics of Computation Laboratory
EPFL, Switzerland

**Lenka Zdeborová**
Statistical Physics of Computation Laboratory
EPFL, Switzerland

## Abstract

We introduce the attention-indexed model (AIM), a theoretical framework for analyzing learning in deep attention layers. Inspired by multi-index models, AIM captures how token-level outputs emerge from layered bilinear interactions over high-dimensional embeddings. Unlike prior tractable attention models, AIM allows full-width key and query matrices, aligning more closely with practical transformers. Using tools from statistical mechanics and random matrix theory, we derive closed-form predictions for Bayes-optimal generalization error and identify sharp phase transitions as a function of sample complexity, model width, and sequence length. We propose a matching approximate message passing algorithm and show that gradient descent can reach optimal performance. AIM offers a solvable playground for understanding learning in self-attention layers, that are key components of modern architectures.

## 1 Introduction

The transformer architecture [1] has transformed machine learning, achieving state-of-the-art results in natural language processing [2, 3], computer vision [4], and beyond. Its core innovation—the self-attention mechanism—enables models to capture long-range dependencies between tokens. Despite their empirical success, transformers remain poorly understood theoretically, especially regarding how data structure, attention bias, and training dynamics interact in finite-sample regimes. While mechanistic interpretability has shed light on trained models, the learning process itself—what is statistically and computationally learnable from limited data—remains unexplained. A common strategy toward progress is to study simplified models in high-dimensional regimes, where the *blessing of dimensionality* [5] can yield tractable characterizations of learning. A key ingredient in this approach is a synthetic data model that captures salient aspects of real-world structure.

A common feature of natural data is that structure arises from pairwise interactions between elements. In language, for example, sentence meaning depends on both the semantics of individual words and their syntactic roles. This can be modeled by assigning each word a *semantic* embedding $x$ and a *contextual* embedding $z$. For instance, in "Max eats chocolate," one level of understanding identifies what each word means (e.g., Max is a person), while another captures grammatical roles (Max is the subject). These two embeddings are linked: contextual meaning emerges by comparing semantic content across words. Similar relational patterns occur in images, molecules, and graphs. The empirical success of transformers suggests that multi-layer self-attention—which performs weighted pairwise comparisons in a hierarchical way—is a natural mechanism for modeling such interactions. To analyze this, we construct a synthetic data model that reflects pairwise structure between tokens.

Theoretical understanding of fully connected neural networks has advanced significantly through the analysis of Gaussian single-index and multi-index models in the high-dimensional limit [6, 7, 8, 9,

39th Conference on Neural Information Processing Systems (NeurIPS 2025).

[10](#), [11](#), [12](#), [13](#), [14](#), [15]. In statistical physics, similar models appear as teacher-student perceptrons [16, 17, 18, 19] or committee machines [20, 21]. These setups typically assume i.i.d. Gaussian inputs, with targets depending on a small number of random projections—"indices"—of the input. They provide a rich theoretical playground for jointly analyzing learning dynamics, generalization, and architectural biases.

Recent work has extended this framework to model key aspects of transformers, introducing the *sequence multi-index* (SMI) model [22, 23, 24]. While insightful, existing SMI models require the width of the key and query matrices to be much smaller than the token embedding dimension—a regime where only narrow attention layers can be analyzed. In contrast, practical transformers typically use key and query widths comparable to the embedding dimension. This motivates our contribution: a high-dimensional yet analyzable model where learnable matrices have extensive rank. We call this the *attention-indexed model*.

**The attention-indexed model (AIM).** We introduce a new class of high-dimensional functions designed to model pairwise relationships between tokens. Analogous to classical multi-index models, the *attention-indexed model* defines outputs $y$ as nonlinear functions of high-dimensional token embeddings $\boldsymbol{x}_a^\top \in \mathbb{R}^d$ for $a = 1, \ldots, T$, arranged as rows of the input sequence of tokens $X_0 \equiv X \in \mathbb{R}^{T \times d}$ for $a = 1, \ldots, T$. We define $L$ *attention indices* $h^{(\ell)} \in \mathbb{R}^{T \times T}$ with components $h_{ab}^{(\ell)}$. The labels $y$ for each input $X \in \mathbb{R}^{T \times d}$ are generated via a general output function $g : \mathbb{R}^{L \times T \times T} \to \mathbb{R}^{T \times T}$

$$h_{ab}^{(\ell)} \equiv \frac{\boldsymbol{x}_a^\top S_\ell \, \boldsymbol{x}_b - \delta_{ab} \operatorname{Tr} S_\ell}{\sqrt{d}} , \qquad y = g\left(\{h^{(\ell)}\}_{\ell=1}^L\right) . \tag{1}$$

Here each $S_\ell \in \mathbb{R}^{d \times d}$ is a learnable matrix. The diagonal mean is subtracted to avoid divergence as $d \to \infty$, ensuring the fluctuations of $h^{(\ell)}$ remain $\mathcal{O}(1)$.

While our theory applies to general rotationally invariant $S_\ell$, a motivating example is when $S_\ell \simeq Q_\ell K_\ell^\top \in \mathbb{R}^{d \times d}$, as in self-attention [1], with key and query matrices $K_\ell, Q_\ell \in \mathbb{R}^{d \times r_\ell}$. We refer to $r_\ell$ as the *width* of the $\ell$th layer; it typically controls the rank of $S_\ell$, though we also consider $r_\ell > d$. For analytical simplicity, we assume tied key and query, $Q_\ell = K_\ell = W_\ell$, so that

$$S_\ell = \frac{1}{\sqrt{r_\ell \, d}} W_\ell W_\ell^\top \in \mathbb{R}^{d \times d} , \quad W_\ell \in \mathbb{R}^{d \times r_\ell} . \tag{2}$$

**Multi-layer attention.** A key instance of the output function $g$ arises when contextual embeddings are built via stacked attention layers. Each layer updates the representation by mixing semantic embeddings through weighted pairwise interactions. Formally, we define the multi-layer attention mechanism as:

$$y = \sigma_\beta\big(H_L(X_{L-1})\big), \qquad X_\ell = \left[c\, \mathbb{I}_T + \sigma_\beta\big(H_\ell(X_{\ell-1})\big)\right] X_{\ell-1} \tag{3}$$

$$H_\ell(X_{\ell-1}) := \frac{1}{\sqrt{d}} X_{\ell-1} S_\ell X_{\ell-1}^\top - \frac{1}{\sqrt{d}} \mathbb{E}_{\mathrm{tr}}[X_{\ell-1} S_\ell X_{\ell-1}^\top] \tag{4}$$

where $X_0 \in \mathbb{R}^{T \times d}$ is the input sequence, $c$ is the residual strength, and $L$ is the number of layers. The function $\sigma : \mathbb{R}^{T \times T} \to \mathbb{R}^{T \times T}$ is typically the softmax with inverse temperature $\beta$; increasing $\beta$ sharpens the focus on dominant pairwise interactions. Finally, the expectation in $H_\ell(X)$ is intended over the input data $X_0$ (empirical average).

A particularly interesting case is the limit of infinite $\beta$, where softmax becomes the *hardmax* function that selects the maximum entry in each row of the attention logits. This can be interpreted as enforcing a sparse, winner-take-all selection among candidate interactions, capturing the case where for each token only a single other token carries the relevant signal. Such mechanisms are especially relevant in symbolic, compositional, or relational reasoning, where outputs depend on identifying a unique best match per query.

This form fits into the attention-indexed framework by choosing the output function $g$ in (1) as:

$$g(\{h^{(\ell)}\}_{\ell=1}^L) = \sigma_\beta\left( B_c^{L-1}(h^{(1)}, \ldots, h^{(L-1)}) \, h^{(L)} \, B_c^{L-1}(h^{(1)}, \ldots, h^{(L-1)})^\top \right), \tag{5}$$

where the recursion is defined by:

$$B_c^0 = \mathbb{I}_T, \quad B_c^\ell = \left[ c\,\mathbb{I}_T + \sigma_\beta \left( B_c^{\ell-1}\, h^{(\ell)}\, B_c^{\ell-1\top} \right) \right] B_c^{\ell-1} \quad \ell = 1, \ldots, L. \tag{6}$$

In particular, with this formulation the map in (4) can be rewritten as:

$$H_\ell(X_{\ell-1}) = \frac{1}{\sqrt{d}}\, X_{\ell-1}\, S_\ell\, X_{\ell-1}^\top - \frac{\operatorname{Tr} S_\ell}{\sqrt{d}}\, B_c^{\ell-1} B_c^{\ell-1\top} = B_c^{\ell-1}\, h^{(\ell)}\, B_c^{\ell-1\top} \tag{7}$$

This formulation also accommodates sequence-level outputs of the form $yX_{L-1}$, corresponding to the output map $gB_c^{L-1}$. Details of the corresponding mappings are given in Appendix B.

The existence of such mapping, from deep attention to a low-dimensional collection of attention indices, is a key contribution: it allows information-theoretic and algorithmic results derived for AIMs to transfer to multi-layer attention. In App. B we extend such mapping to multi-head and seq2seq variants. Full Transformer blocks (e.g., ViT/LLMs) interleave multi-head self-attention with token-wise MLPs, and are commonly trained in an auto-regressive fashion; our supervised setup isolates the attention learning problem, to which our results apply directly.

**Our contributions.** We initiate a theoretical study of learning from data generated by the attention-indexed model, without restrictions on the width of the attention matrices $S_\ell$, in the high-dimensional limit. Specifically, we consider large embedding dimension $d$, with attention widths $r_\ell$ (or ranks of $S_\ell$) scaling proportionally with $d$, while keeping sequence length $T$ and depth $L$ finite. The number of samples $n$ scales quadratically with dimension to span the regime where the optimal test error changes from as bad as a random guess to zero:

$$d \to \infty, \quad \alpha \equiv \frac{n}{d^2} = \Theta(1), \quad \rho_\ell \equiv \frac{r_\ell}{d} = \Theta(1), \quad T = \Theta(1), \quad L = \Theta(1). \tag{8}$$

In this limit, for Gaussian i.i.d. inputs and suitable priors over $S_\ell$, we analyze the Bayes-optimal estimator—that is, the posterior mean predictor. Concentration of measure allows key observables—such as the test error—to become deterministic, enabling a description in terms of low-dimensional fixed-point equations. We derive those equations using tools from statistical mechanics and random matrix theory. Beyond analytical tractability, this asymptotic setting captures relevant scalings, where model and data dimensions grow together.

Our analysis uncovers phase transitions in learnability as a function of sample complexity $\alpha$, sequence length $T$, and attention width $\rho$. We show that attention mechanisms with extensive width can efficiently recover latent structure, particularly when the target model exhibits sparsity. We further study the role of the inverse temperature parameter $\beta$, contrasting the smooth regime $0 < \beta < +\infty$ with the hard assignment case $\beta = +\infty$, where the task reduces to discrete token association. We derive an approximate message passing (AMP) algorithm that matches Bayes-optimal performance in the studied setting. We also show empirically that an averaged version of gradient descent on a natural loss achieves similar optimality, aligning theory with practical training dynamics.

**Further related work.** The attention-indexed model (AIM) is motivated by a generative perspective, capturing how structured token-level outputs arise from layered bilinear interactions between high-dimensional embeddings—mirroring attention computations in transformers. The idea of modeling learning through such structured synthetic data dates back to the teacher–student setting in the perceptron literature [16, 17, 20], and more recently to single-index and multi-index models [6, 7, 8, 9, 10, 11, 12, 13, 14, 15].

While many theoretical studies explore simplified transformer variants, most do not rely on or benefit from the high-dimensional limit. These include works that analyze one-layer attention under finite embedding dimension [25, 26, 27, 28, 29], or study training dynamics in the linear, kernel, or random feature regimes [30, 31, 32]. Others use infinite-width approximations without access to generalization error [33, 34]. By contrast, theoretical analysis of *nonlinear* attention layers with *trainable* key and query matrices in the limit of high embedding dimension—together with sharp control of generalization—is less explored. As far as we are aware, only a few works address this regime [23, 22, 35, 36], and they all assume attention matrices of *finite* width.

Methodologically, our approach builds on techniques from high-dimensional multi-index models, particularly those developed in [21, 37], and their recent generalizations to sequence learning with multiple low-width self-attention layers [35]. The main technical challenge addressed in this paper is

extending these tools to the case where the width $r$ of the attention matrices scales proportionally with the embedding dimension—i.e., the extensive-width regime—going beyond the key limitations of prior analyses.

To tackle this, we leverage recent results on the ellipsoid fitting problem [38, 39] and its connection to two-layer neural networks with quadratic activations and extensive width [40, 41]. Remarkably, the linear AIM model with $T = L = 1$ is mathematically equivalent to such quadratic networks, allowing us to adopt these methods. We generalize this connection to arbitrary $T, L$. This is enabled by a central conceptual tool, the AIM index, which disentangles the complexity of deep attention models. It allows us to split the problem into two subproblems: (i) how structure propagates across layers and tokens, and (ii) how attention matrices are learned from those structures. This separation is crucial in extending the theory to multiple layers and tokens.

Finally, we note that we focus here on the tied case $Q = K$ for clarity. The untied setting $Q \neq K$ is amenable to similar analysis following [42], and we leave its treatment for future work.

## 2  Setting

We consider a dataset $\mathcal{D} = \{y^\mu, X_0^\mu\}$ of $n$ samples indexed by $\mu$, where $X_0^\mu = \{\boldsymbol{x}_a^{\mu\top}\}_{a=1}^T \in \mathbb{R}^{T \times d}$ has rows $\boldsymbol{x}_a^{\mu\top}$. Each sample consists of the embeddings of $T$ tokens $\boldsymbol{x}_a^{\mu\top} \in \mathbb{R}^d$, taken as standard Gaussian $\boldsymbol{x}_a^{\mu\top} \sim \mathcal{N}(0, \mathbb{I}_d)$ and of $T \times T$ matching output matrices $y^\mu$ encoding pair-wise information on the original tokens. We stress that the Gaussian assumption for the data can be relaxed in the same spirit as in [41, Assumption 2.2].

We generate $y^\mu$ using an attention-indexed model as given in (1) with matrices $\{S_\ell^*\}_{\ell=1,\dots,L}$ that are symmetric and extracted independently from a rotationally invariant ensemble $P_S(S) = P_S(O^\top S O)$ for any $d \times d$ rotation matrix $O$. We fix the normalizations such that $\mathbb{E}_{P_S}[\operatorname{Tr} S] = \kappa_1 d$ and $\mathbb{E}_{P_S}[\operatorname{Tr} S^2] = \kappa_2 d$ and with $\kappa_1, \kappa_2 = \mathcal{O}(1)$. We assume that the empirical spectral distribution of $S \sim P_S$ converges to a a distribution $\mu_S$ for $d \to +\infty$. This setting can be relaxed in several directions, allowing for different prior distributions $P_S^{(\ell)}$ for different layers, as well as considering non-symmetric matrices [43].

We consider the Bayes-optimal (BO) learning setting: the statistician knows the generative process of the dataset, i.e. the non-linearity $g$ in (1) and the prior distribution $P_S$, and observes a dataset $\mathcal{D}$ but not the specific set of weights $\{S_\ell^*\}_{\ell=1,\dots,L}$ used to generate said dataset. The task is then to optimally estimate either the weights $S^*$ (estimation task), i.e. find the estimator $\hat{S}(\mathcal{D})$ that minimizes

$$\mathcal{E}_{\text{est}}(\hat{S}) = \mathbb{E}_{\mathcal{D}, S^*} \frac{1}{d} \sum_{\ell=1}^L ||\hat{S}(\mathcal{D})_\ell - S_\ell^*||_F^2 \,, \tag{9}$$

or the label associated to a new input sample $X_{\text{new}}$ (generalization task), i.e. find the estimator $\hat{y}(\boldsymbol{x}, \mathcal{D})$ that minimizes

$$\mathcal{E}_{\text{gen}}(\hat{y}) = \mathbb{E}_{\mathcal{D}, S^*} \mathbb{E}_{y_{\text{new}}, X_{\text{new}}} ||\hat{y}(X_{\text{new}}, \mathcal{D}) - y_{\text{new}}||_F^2 \,, \tag{10}$$

where $(y_{\text{new}}, X_{\text{new}})$ is a new label-sample pair generated with the weights $S^*$. We will call the error achieved by the optimal estimators, respectively, the BO estimation error and BO generalization error.

Both BO estimators can be computed from the knowledge of the posterior distribution, i.e. the probability that a given set of weights $S$ was used to generate the observed dataset

$$P(S_1, ..., S_L | \mathcal{D}) = \frac{1}{\mathcal{Z}(\mathcal{D})} \prod_{\ell=1}^L P_S(S_\ell) \prod_{\mu=1}^n \delta\left(y^\mu - g\left(h^{(1)}(S_1, \boldsymbol{x}^\mu), ..., h^{(L)}(S_L, \boldsymbol{x}^\mu)\right)\right) \,, \tag{11}$$

where the attention indices $h^{(\ell)} \in \mathbb{R}^{T \times T}$ were defined in (1) and $\mathcal{Z}(\mathcal{D})$ is a normalization factor. The BO estimator with respect to the estimation error is the mean of the posterior distribution, while BO estimator with respect to the generalization error is the mean of the predicted label under the posterior distribution, i.e.

$$\hat{S} = \mathbb{E}_{S \sim P(S|\mathcal{D})}[S] \,, \qquad \hat{y}(\boldsymbol{x}) = \mathbb{E}_{S \sim P(S|\mathcal{D})}\left[g\left(h^{(1)}(S_1, \boldsymbol{x}), ..., h^{(L)}(S_L, \boldsymbol{x})\right)\right] \,. \tag{12}$$

Under the high-dimensional limit (8), we will show that the estimation/generalization error achieved by the BO estimators are characterized through the *overlaps* $q, Q$ defined as

$$q_{\ell k} = \frac{1}{d} \mathbb{E}_{S \sim P(S|\mathcal{D})}\left[\operatorname{Tr} S_\ell S_k^*\right] \,, \qquad Q_{\ell k} = \frac{1}{d} \mathbb{E}_{S \sim P_S}\left[\operatorname{Tr} S_\ell^* S_k^*\right] \,. \tag{13}$$

# 3 Statistical and computational limits for AIMs

In this Section we provide results for the information-theoretical and computational limits on attention-indexed models in a very general setting, which we then specify to the attention models in Section 4.

In order to state our results let us define, for two $L \times L$ symmetric positive-definite matrix $q, \hat{q} \in \mathbb{S}_L^+$

$$
\mathcal{Z}_{\text{in}}(\{Y_\ell\}_{\ell=1}^L; \hat{q}) = \int \left[\prod_{\ell=1}^L dS_\ell \, P_S(S_\ell)\right] \exp\left[-\frac{d}{4} \sum_{\ell,k=1}^L \hat{q}_{\ell k} \operatorname{Tr}(S_\ell S_k) + \frac{d}{2} \sum_{\ell,k=1}^L \sqrt{\hat{q}}_{\ell k} \operatorname{Tr}(Y_\ell S_k)\right].
$$

$$
I_{\text{in}}(\hat{q}) = \lim_{d\to\infty} \frac{1}{d^2} \int DY_1 \dots DY_L \, \mathcal{Z}_{\text{in}}(Y_1, \dots, Y_L; \hat{q}) \log \mathcal{Z}_{\text{in}}(Y_1, \dots, Y_L; \hat{q})
$$

(14)

where $DY$ stands for integration over a $\text{GOE}(d)$ (Wigner) matrix $Y$, and

$$
\mathcal{Z}_{\text{out}}(y, \omega, V) = \int \left[\prod_{a \leq b}^T d^L h_{ab} \, \mathcal{N}(h_{ab}; \omega_{ab}; V_{ab})\right] \delta\left(y - g(\{h_{ab}^{(\ell)}/\sqrt{2 - \delta_{ab}}\}_{\ell=1}^L)\right)
$$

$$
I_{\text{out}}(q) = \int \prod_{a,b=1}^T dy_{ab} \int \mathcal{D}\eta_1 \dots \mathcal{D}\eta_L \, \mathcal{Z}_{\text{out}}(y, \omega, V) \log \mathcal{Z}_{\text{out}}(y, \omega, V)
$$

(15)

$$
\text{with:} \quad \omega_{ab}^{(\ell)} = \sum_{k=1}^L \sqrt{2q}_{\ell k} \, \eta_{ab}^{(k)}, \qquad V^{(\ell k)} = 2(Q_{\ell k} - q_{\ell k})
$$

where $\mathcal{D}\eta$ stands for integration over a $L \times T \times T$ tensor symmetric in the token indices and with independent entries $\mathcal{N}(0, 1)$. Moreover, let us define

$$
g_{\text{out}}(y, \omega, V) = \partial_\omega \ln \mathcal{Z}_{\text{out}}(y, \omega, V), \qquad g_{\text{in}}(Y|\hat{q}) = \partial_{\hat{q}^{1/2} Y} \ln \mathcal{Z}_{\text{in}}(\hat{q}).
$$

(16)

Our first result provides a description of the error in the high dimensional $d \to \infty$ limit, with finite sample complexity $\alpha = n/d^2$, number of tokens $T$ and number of layers $L$.

**Result 3.1** (Performance of information-theoretically optimal estimators). *Consider the extremization problem*

$$
\inf_{\hat{q} \in \mathbb{S}_L^+} \sup_{q \in \mathbb{S}_L^+} \left\{-\frac{\operatorname{Tr} q\hat{q}}{4} + I_{\text{in}}(\hat{q}) + \alpha I_{\text{out}}(q)\right\}
$$

(17)

*where $I_{\text{in}}(\hat{q})$ and $I_{\text{out}}(q)$ are defined in (14) and (15), Call $(q^*, \hat{q}^*)$ the global extremizer of (17). Then, in the high dimensional limit (8), the BO estimation error is given by*

$$
\lim_{d\to\infty} \mathcal{E}_{\text{est}}(\hat{S}_{\text{BO}}) = \|Q - q^*\|_F^2,
$$

(18)

*while the BO generalization error is given by*

$$
\mathcal{E}_{\text{gen}} = \mathbb{E}_{\eta,\xi} \left\| g\left(\left\{\frac{\omega_{ab}^{(\ell)}}{\sqrt{2 - \delta_{ab}}}\right\}_{a \leq b, \ell=1}^{T,L}\right) - g\left(\left\{\frac{h(\omega^{(\ell)}, V^{(\ell k)})_{ab}}{\sqrt{2 - \delta_{ab}}}\right\}_{a \leq b, \ell=1}^{T,L}\right)\right\|_F^2,
$$

(19)

*where $\xi$ a $L \times T \times T$ tensor symmetric in the token indices and with independent entries $\mathcal{N}(0, 1)$ and $\omega, V, \eta$ are defined in (15). Finally $h(\omega^{(\ell)}, V^{(\ell k)}) = \omega^{(\ell)} + \sum_{k=1}^L \sqrt{V}_{\ell k} \xi^{(k)}$.*

These expressions provide a prediction for the information-theoretically optimal estimation and generalization errors, and as such they constitute a sharp information theoretical bound on the performance of any algorithm. We derive Result 3.1 in Appendix C using the heuristic replica method, but believe that a rigorous treatment is possible by adapting [41] to the case of multiple attention indices $L > 1$ and multiple tokens $T > 1$.

In our next result we provide an efficient polynomial-time approximate message passing (AMP) algorithm that in the high-dimensional limit saturates this bound under the condition that there is a unique local extremizer of (17) (if multiple extermizers are present, computational-to-statistical gaps may arise [44]).

**Algorithm 1:** AMP

---

**Result:** The estimators $\hat{S}_\ell$
**Input:** Observations $y^\mu \in \mathbb{R}^{T \times T}$ and "sensing matrices"
$Z^\mu_{ij,ab} \equiv (\boldsymbol{x}^\mu_{i,a}\boldsymbol{x}^\mu_{j,b} + \boldsymbol{x}^\mu_{j,a}\boldsymbol{x}^\mu_{i,b} - 2\delta_{ij}\delta_{ab})/\sqrt{2d(1+\delta_{ab})} \in \mathbb{R}$;
*Initialize* $\hat{S}^{t=0}_\ell \sim P_S$ and $\hat{C}^{t=0} = 2(\kappa_2 - \kappa_1^2)\mathbb{I}_L$;
**while** *not converging* **do**

    • *Estimation of the variance and mean of* $\mathrm{Tr}[Z^\mu_{ab}S_\ell]$;

    $V^t = 2\hat{C}^t$    and    $\omega^t_{\mu,\,ab} = \mathrm{Tr}[Z^\mu_{ab}\hat{S}^t] - (1-\delta_{0t})g_{\mathrm{out}}(y^\mu,\omega^{t-1}_\mu,V^{t-1})_{ab}V^t \in \mathbb{R}^L$ ;

    • *Variance and mean of* $S_\ell$ *estimated from the "output" observations*;

    $\hat{q}^t_{\ell k} = \dfrac{4\alpha}{n}\displaystyle\sum_{\mu,a\leq b}^{n,T,T} g_{\mathrm{out}}(y^\mu,\omega^t_\mu,V^t)^{(\ell)}_{ab}\,g_{\mathrm{out}}(y^\mu,\omega^t_\mu,V^t)^{(k)}_{ab}$    and

    $R^t_{ij} = \hat{S}^t_{ij} + (\hat{q}^t)^{-1}\dfrac{2}{d}\displaystyle\sum_{\mu,a\leq b}^{n,T,T} g_{\mathrm{out}}(y^\mu,\omega^t_\mu,V^t)_{ab}Z^\mu_{ij,ab} \in \mathbb{R}^L$;

    • *Update of the estimation of* $S$ *with the "input" information*;

    $\hat{S}^{t+1}_\ell = g_{\mathrm{in}}\left(R^t,\hat{q}^t\right)_\ell$      and      $\hat{C}^{t+1}_{\ell k} = \dfrac{1}{d^2}\nabla_{R_k}\cdot g_{\mathrm{in}}\left(R^t,\hat{q}^t\right)_\ell$;

    $t = t+1$;

**end**

---

**Result 3.2** (State evolution of AMP). *Call $\hat{S}^t_\ell$ the time-$t$ iterate of the AMP algorithm 1. In the high dimensional limit (8) of large $d$ and for a finite number of iterations, we have that*

$$\frac{1}{d}\mathrm{Tr}[\hat{S}^t_\ell \hat{S}^t_k] \to q^t_{\ell k}, \qquad\qquad \frac{1}{d}\mathrm{Tr}[\hat{S}^t_\ell S^*_k] \to q^t_{\ell k}, \qquad\qquad (20)$$

*where*

$$\hat{q}^{t+1}_{\ell k} = 4\alpha\mathbb{E}_{\xi,\eta}\sum_{a\leq b}^{T} g_{\mathrm{out}}\left(g\left(\left\{\frac{h(\omega^t,V^t)_{ab}}{\sqrt{2-\delta_{ab}}}\right\}\right),\omega^t,V^t\right)^{(\ell)}_{ab}$$

$$\times g_{\mathrm{out}}\left(g\left(\left\{\frac{h(\omega^t,V^t)_{ab}}{\sqrt{2-\delta_{ab}}}\right\}\right),\omega^t,V^t\right)^{(k)}_{ab}, \qquad\qquad (21)$$

$$q^{t+1}_{\ell k} = \lim_{d\to+\infty}\frac{1}{d}\mathbb{E}_{S,Y}\mathrm{Tr}\left[g_{\mathrm{in}}(Y(S,\hat{q}^{t+1}),\hat{q}^{t+1})_\ell\, g_{\mathrm{in}}(Y(S,\hat{q}^{t+1}),\hat{q}^{t+1})_k\right],$$

*with $V^t = 2(Q-q^t)$ and $\omega^t_\ell = \sum_{k=1}^L(\sqrt{2q^t})_{\ell k}\,\eta_k$, where $(a,b)$ are token and $(\ell,k)$ layer indices and with $\eta_\ell$ a standard Gaussian in every component. $h(\omega^t,V^t)$, $\eta$ and $\xi$ are defined as in Result 3.1. Finally we have $Y(S,\Delta)_\ell = S_\ell + \sum_{m=1}^L\sqrt{\Delta}_{\ell m}\Xi_m$, all $\Xi_m$ are $\mathrm{GOE}(d)$ and all $S_\ell \sim P_S$.*

Result 3.2 (which we derive in Appendix C) is a classic statement in the theory of AMP, that here we adapt to take into account multiple attention indices and multiple tokens. We remark that even though the denoiser $g_{\mathrm{in}}$ in (16) is a complicated non-separable function, state evolution still holds [45, 46]. Additionally, we show in Appendix C that the fixed point of (21) are the extremisers of (17). The AMP algorithm 1 is thus both a tool that gives us theoretical guarantees though (21) and a practical algorithm that can be efficiently implemented and run on a machine. We discuss the implementation in Appendix C.

The prior-related elements $I_{\mathrm{in}}$ and $g_{\mathrm{in}}$ of (14) and (21) require discussion. Both of them involve integrals in dimension $\mathcal{O}(Ld^2)$ whose large $d$ asymptotics is highly non-trivial.

This is due to the prior $P_S$ on the weights being non-separable, and crucially to the coupling between weights of different layers caused by the non-diagonal matrix $\hat{q}$ in (14) and (21). We remark that for a generic number of layers $L > 1$, the evaluation of these two equations is equivalent to the evaluation

in the high-dimensional limit of the free entropy and of the Bayes-optimal estimator for the following $L$-wise matrix denoising problem

$$Y_\ell = S_\ell + \sum_{m=1}^{L} \sqrt{\Delta}_{\ell m} \Xi_m \,, \tag{22}$$

where from a heavily correlated set of $L$ noisy observations $Y_\ell \in \mathbb{R}^{d \times d}$, $\ell = 1, \ldots, L$, one needs to estimate back the $L$ independent matrices $S_m \sim P_S$. Here $\Delta \in \mathbb{S}_L^+$ is an $L$-dimensional symmetric positive-definite matrix acting as a noise-to-signal ratio, and $\Xi_\ell$ are independent $\text{GOE}(d)$ matrices acting as noise. To the best of our knowledge, for $L > 1$ and non-diagonal $\Delta$ this is still a challenging open problem (in case of diagonal $\Delta$, the problem factorizes in $L$ independent matrix denoising problems). In practice, one can approximate rotationally-invariant matrix denoisers by low-degree spectral polynomials, following [47]; this provides a practical route to AMP for multi-layer $\hat{q}$ when the exact HCIZ-based denoiser is unavailable [48].

For $L = 1$, the problem (22) reduces to Bayes-optimal denoising of a rotationally invariant matrix [40, 43]. We start by remarking that for $L = 1$ the quantities $Q, q, \hat{q}$ governing the BO performance are scalar.

For Result (3.1) at leading order in $d \to \infty$ we have

$$I_{\text{in}}(\hat{q}) = \frac{Q}{4}\hat{q} - \frac{1}{4}\log\hat{q} - \frac{1}{2}\Sigma(\mu_{1/\hat{q}}) - \frac{1}{8} \quad \text{where} \quad \Sigma(\mu) = \mathbb{E}_{x,y \sim \mu} \log|x - y| \,, \tag{23}$$

and for the AMP algorithm Result (3.2) we have

$$g_{\text{in},L=1}(X, \Delta) = O^\top f_{\text{RIE}}(\Lambda, \Delta) O \quad \text{where} \quad f_{\text{RIE}}(\Lambda, \Delta)_i = \Lambda_i - 2\Delta \int \frac{d\mu_\Delta(t)}{\Lambda_i - t} \,, \tag{24}$$

where $Q = \kappa_2$, $\mu_\Delta$ is the asymptotic spectral density of a matrix $X = S + \sqrt{\Delta}Z$ with $S \sim P_S$ and $Z \sim \text{GOE}(d)$, and $X = O^\top \Lambda O$ is the eigen-decomposition of $X$ (see [40] for details).

Similarly, at leading order in $d \to \infty$ we have

$$\frac{1}{d^2}\nabla_X \cdot g_{\text{in},L=1}(X, \Delta) = \Delta - \frac{4\pi^2\Delta^2}{3}\int d\mu_\Delta(t)^3 \,. \tag{25}$$

Finally, the state evolution equation for $q$ in (21) reads

$$q = Q - \frac{1}{\hat{q}} + \frac{4\pi^2}{3\hat{q}^2}\int dt\,[\mu_{1/\hat{q}}(t)]^3 \,. \tag{26}$$

## 4 Results for single-layer attention

We now apply our general results to the single-layer ($L = 1$) tied-attention model

$$y_{ab} = \sigma_\beta\left(\frac{\boldsymbol{x}_a^\top S\,\boldsymbol{x}_b - \delta_{ab}\operatorname{Tr}S}{\sqrt{d}}\right) = \sigma_\beta\left(\frac{\frac{1}{\sqrt{rd}}\boldsymbol{x}_a^\top WW^\top\,\boldsymbol{x}_b - \delta_{ab}\operatorname{Tr}\left(\frac{1}{\sqrt{rd}}WW^\top\right)}{\sqrt{d}}\right) \tag{27}$$

where we parametrized the weight matrix $S$ as a tied-attention with extensive-width $r = \rho d$ and $W \in \mathbb{R}^{d \times r}$ has independent entries $W_{ij} \sim \mathcal{N}(0, 1)$. For the activation, we consider the case of Hardmax $\sigma_{\text{hard}}$ and Softmax $\sigma_{\text{soft}}$, both applied row-wise in (27):

$$\sigma_{\text{hard}}(z_1 \ldots z_T)_i = \delta(i = \arg\max_j x_j)\,, \quad \text{and} \quad \sigma_{\text{soft}}(z_1 \ldots z_T)_i = \frac{e^{\beta z_i}}{\sum_{j=1}^T e^{\beta z_j}}\,. \tag{28}$$

We stress that both these tasks are well-defined only for $T \geq 2$, as the $T = 1$ the output of both activations equals 1 regardless of the input. As discussed in the introduction, the model with hardmax provides an interesting token-association task.

**Hardmax target.** The BO treatment of the hardmax activation for generic number of tokens $T$ is challenging due to the complex form of the state equation for $\hat{q}$ (21). We provide an explicit solution in the $T = 2$ case.

**Result 4.1** (Bayes-optimal errors for hardmax tied-attention, $T = 2$). *Consider the model* (27) *with hardmax activation. In the high-dimensional limit* (8), *the asymptotic BO estimation and generalization errors are given by* (18) *and* (19), *where* $(q, \hat{q})$ *is the solution of* (26) *with* $Q = 1 + \rho$ *and*

$$g_{\text{out}}(y, \omega, V)_{ab} = \frac{1}{\sqrt{6(Q-q)}} \frac{\phi(k_1, k_2, c)}{\Phi(k_1, k_2, c)} \begin{pmatrix} \sqrt{2}s_1 & -(s_1 + s_2) \\ -(s_1 + s_2) & \sqrt{2}s_2 \end{pmatrix}_{ab}, \qquad (29)$$

*where* $\phi(k_1, k_2, c)$ *is the p.d.f. of a bi-variate Gaussian with zero mean, variances* $1/(1-c^2)$ *and covariance* $c/(1-c^2)$, *and* $\Phi(k_1, k_2, c)$ *is its c.d.f (see Appendix* A). *Moreover,* $s_a = 2y_{aa} - 1$, $k_a = s_a(\sqrt{2}\omega_{aa} - \omega_{12})/\sqrt{6(Q-q)}$, $c = s_1 s_2 / 3$ *and* $\omega_{ab} = \sqrt{2q}\,\eta_{ab}$.

We detail the derivation of Result 4.1 in App. C. We plot the estimation error given in Result (4.1) in Figure 1 left, for several values of the attention width ratio $\rho$, comparing with runs of the associated AMP Algorithm 1 at size $d = 100$.

We observe that for all finite $\alpha$ the estimation error is strictly positive, and that it approaches zero as $\alpha$ grows with rate compatible with $\mathcal{O}(1/\alpha)$. Moreover, as soon as $\alpha > 0$, we observe that the estimation error is smaller than 1, i.e. the value achieved in the absence of data. Intuitively, the hardmax output function enforces discrete token association per row. This is akin to a multiclass classification target, which explains the lack of a finite strong-recovery threshold and the observed power-law decay at large $\alpha$ (Fig. 1, left).

In the limit of small width Result 4.1 simplifies. Notice that in this limit the correct sample scale is given by $\bar{\alpha} = \alpha/\rho = n/(dr)$, as the matrix to infer is not extensive-width anymore. In this limit there appears a so–called weak recovery threshold, a value of sample complexity below which the estimator reaches the same performance as if there were no data. We characterize it as follows.

**Corollary 4.2** (Small width limit for hardmax activation). *Consider the model* (27) *with hardmax activation and* $T = 2$. *In the high-dimensional limit* (8), *the equation for* $q$ *of Result* 3.1 *simplifies to* $q = [\max(1 - t, 0)]^2$ *under the rescaling* $\alpha = \bar{\alpha}\rho$ *and* $\hat{q} = t\rho$. *In particular, the BO error is the same as that of the data-less estimator for all* $\bar{\alpha} < \bar{\alpha}_{\text{weak}}^{\text{hardmax}}$ *where*

$$\bar{\alpha}_{\text{weak}}^{\text{hardmax}} = \frac{1}{4\mathbb{E}_{y,\omega}\left[\sum_{a \leq b}^{T=2} g_{\text{out}}(y(\omega, V), \omega, V)_{ab}^{\otimes 2}\right]_{q=0, Q=1}} \approx 0.563. \qquad (30)$$

Corollary 4.2 follows by combining Result 4.2 with the small-width analysis of [40]. We remark that (30) holds for all activation (using the appropriate $g_{\text{out}}$ function) and all values of $T \geq 2$. We plot the analytical prediction for the BO estimation error given in Corollary 4.2 in Figure 1 right.

**Softmax target.** We now discuss the target function that uses a softmax non-linearity (28). This choice of activation allows for an analytic treatment for any number of tokens $T \geq 2$, and any finite value of the softmax inverse temperature $\beta \in \mathbb{R}_+$.

**Result 4.3** (Bayes-optimal errors for softmax tied-attention, $T \geq 2$). *Consider the model* (27) *with softmax activation,* $T \geq 2$ *and inverse temperature* $\beta \in \mathbb{R}_+$. *In the high-dimensional limit* (8), *the asymptotic BO estimation and generalization errors are given by* (18) *and* (19), *where* $(q, \hat{q}) \in \mathbb{R}_+^2$ *is the solution of* (26), $Q = 1 + \rho$ *and* $\hat{q} = \alpha(T^2 + T - 2)/(Q - q)$.

We plot the BO estimation error given in Result 4.3 in Figure 2 left, and observe that contrary to the hardmax case, the BO estimation error vanishes at a finite value of $\alpha$ (the so-called strong recovery threshold). Interestingly, the BO errors given in Result (4.3) is independent of the value of the inverse temperature $\beta$ and reduces to the case of a single-token model with linear activation [40], modulo a rescaling of the sample ratio $\alpha$ to $2\alpha/(T^2 + T - 2)$ (notice that the rescaling is not just given by the total number unordered couples of tokens $T(T + 1)/2$, as it would be in the case of a multi-token case with bijective activation, see App. C). The softmax activation is almost invertible, meaning that given the output, the input is fully determined apart for a common additive shift (acting as a noise correlated with the data), and is additionally constrained by the symmetry of the attention matrix. Result (4.3) precisely quantifies the amount of samples required to estimate this undetermined shift. More precisely, fix a given estimation error. Then, achieving this error with the BO estimator in the softmax case with $T \geq 2$ requires a factor $1 + 2/(T(T + 1))$ more samples than the case of a fully bijective activation.

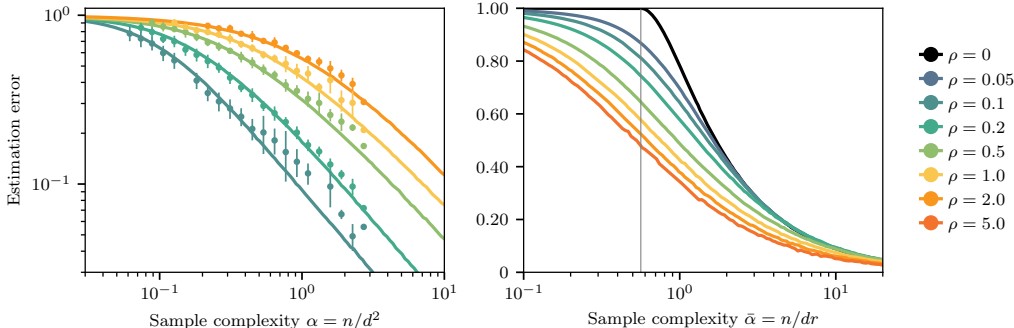

Figure 1: **(Left)** The Bayes optimal-error for the single-layer attention-indexed model with $T = 2$ tokens and hardmax activation for and several values of the width ratio $\rho$ (Result (4.1)). The log-log scale highlights a large $\alpha$ power-law decay of the BO estimation error, strikingly different from the softmax behaviour (see Figure 2). We also plot the corresponding errors achieved by the AMP algorithm (dots) at $d = 100$, averaged over 16 realizations of the data and teacher weights. Error bars are computed with respect to the mean. We find a good agreement even for such a moderate size. **(Right)** Focus on the small width Bayes-optimal error case (Corollary (4.2)) of the same model. We rescale the sample complexity to $\bar{\alpha} = \alpha/\rho$, and we highlight the theoretical prediction of the weak recovery thresholds (gray vertical line).

On the other hand, we remark that the AMP algorithm for the softmax activation at $T \geq 2$ *is not* a simple rescaling of the AMP for the single-token linear-activation case given in [40]. The AMP output function $g_{\text{out}}$ is given by

$$g_{\text{out},ab} = -\frac{\tau_{ab}}{T^2 V}\Big[\sum_{c \leq d}^{T}[\tau_{cd}^2 \phi_{Tc} - \tau_{cd}\omega_{cd}] + \sum_{c \leq d}^{T-1}\tau_{cd}^2\phi_{cd}\Big] + \frac{\tau_{ab}\phi_{Ta} - \omega_{ab} + (1 - \delta_{bT})\phi_{ab}\tau_{ab}}{V} , \quad (31)$$

where $\tau_{ab} = \sqrt{2 - \delta_{ab}}$, $\phi(y)_{ab} = \beta^{-1}\log(y_{ab}/y_{aT})$ and $\omega, V$ defined in (15). Thus, AMP processes the data in a non-trivial, optimal way to perform this effective inversion of the softmax activation. We plot experiments for AMP at $d = 100$ in the $T = 2, 3$ case in Figure 2 right (purple and blue dots, to be compared with the prediction of Result (4.3) given by the black line), and observe a nice agreement. We also remark that while the BO performance is independent of the inverse temperature $\beta$, as long as it is finite, again AMP output function is not.

Thanks to the mentioned reduction, one can transfer directly several results from [40] to the case of softmax tied-attention, including an explicit prediction for the strong recovery threshold (the value of $\alpha$ after which the BO error is zero), the slope of the error at strong recovery, and the small-width and large-width limits (see App. D). In particular, the strong recovery threshold satisfies

$$\alpha_{\text{recovery}}^{\text{softmax}} = \frac{2}{T^2 + T - 2}\begin{cases} \rho - \rho^2/2 & \text{if} \quad 0 < \rho < 1 \\ 1/2 & \text{if} \quad \rho \geq 1 \end{cases} . \quad (32)$$

We remark again that this threshold does not coincide with the naive counting argument, which would give a factor $\frac{T(T+1)}{2}$ at denominator instead. In particular, the $2/(T^2+T-2) = \frac{T(T+1)}{2} - 1$ factor reflects the near-invertibility of the row-wise softmax under a global row-shift, modulo the symmetry constraint; cf. App. D. Finally, contrarily to hardmax (which implements a discrete winner-takes-all assignment and exhibits power-law error decay), softmax behaves like a smooth regression target with a finite strong-recovery threshold (Eq. (32)), see Fig. 1 vs Fig. 2.

We finally remark that our analysis keeps $T, L$ finite while $d, n \to \infty$ with $n/d^2 = \alpha = \Theta(1)$. The predictions remain accurate as long as $T$ grows much slower than $d$; the curves obtained after the $T$-dependent rescaling of $\alpha$ (Sec. 4) remain valid for very large $T$ provided $T << d$.

Finally, we consider the performance of gradient descent minimizing the loss

$$\mathcal{L}(W) = \sum_{\mu=1}^{n}\sum_{a,b=1}^{T}\left(y_{ab}^{\mu} - \sigma_\beta\left(\frac{\boldsymbol{x}_a^{\mu\top}WW^\top\,\boldsymbol{x}_b^\mu - \delta_{ab}\operatorname{Tr}WW^\top}{\sqrt{r}\,d}\right)\right)^2 , \quad (33)$$

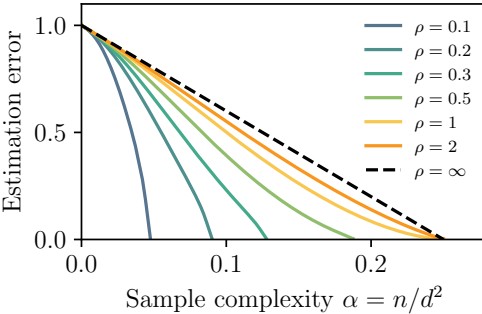 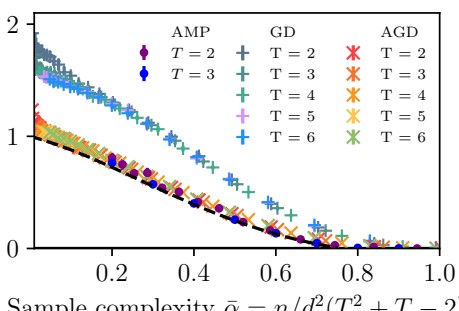

Figure 2: **(Left)** Illustration of the Bayes-optimal estimation error for the softmax tied-attention model (Result 4.3), eq. (27) for any $0 < \beta < +\infty$ and $T = 2$ tokens, and for several values of the attention width ratio $\rho = r/d$. The model reaches zero BO error at finite $\alpha$ depending on $\rho$ (eq. (32)). **(Right)** We show, in black dashed lines the theoretical prediction of the BO estimation error computed for the sample complexity rescaled by the number of tokens $\alpha/(T^2 + T - 2)$ and $\rho = 0.5$. We show the performance of the corresponding AMP algorithm for $T = 2, 3$ tokens, correctly achieving the BO error. We also compare the BO performance with those of Adam GD and its averaged version AGD with $d = 100$. We average each numerical experiment (GD,AGD,AMP) over 16 realizations of the data and teacher weights. Error bars are the standard deviation on the mean.

with training set generated by Eq. (27) (we optimize using the ADAM optimizer [49]). In line with previous work [40, 42], we also consider the Averaged GD (AGD) estimator given by

$$\hat{S}_{\text{GD,avg}} = \frac{1}{M} \sum_{m=1}^{M} \frac{W_m^{\text{final}}(W_m^{\text{final}})^\top}{\sqrt{rd}} \,, \tag{34}$$

where we average over $M$ initial matrices $W_m^{(0)}$, and $W_m^{\text{final}}$ is the corresponding set of weights at convergence. We plot the results of our numerical experiments at $d = 200$ for both GD and AGD in Figure 2 right. As already observed in [40, 42], AGD reaches performances compatible with the BO estimation error, while GD has worse error. We remark that both variants seem to achieve perfect recovery at the BO threshold (32). This phenomenon, at this point well documented within this class of models, is still not understood.

## 5   Limitations

The attention-indexed model introduced in this work and its high-dimensional analysis provide promising stepping stone to the analysis of learning in multi-layer attention. We, however, so far only analyzed the Bayes-optimal performance and the associated AMP algorithm in the single layer case, gradient descent was only explored numerically and clearly its theoretical analysis in this class of models is an interesting topic for future work. While the concept of attention-indexed model captures multi-layer attention networks, the matrix denoiser (22) needed to study the optimal performance is a challenging open problem for random matrix theory and thus the analysis of the model for more than one layer remains open. While out results rely on exact statistical physics tools, mathematically rigorous proof of the obtained results for $T \geq 2$ is an open problem. Future work should also explore the inclusion of multiple heads and MLP layers to mimic yet closer practicalities of the current transformer architectures. The structure of the input data considered in this paper is very simple, future work should also include studied of how the performance depends on the input structure.

## Acknowledgments and Disclosure of Funding

We thank Antoine Maillard, Florent Krzakala, Hugo Cui and Yizhou Xu for insightful discussions related to this work. We acknowledge funding from the Swiss National Science Foundation grants SNSF SMArtNet (grant number 212049).

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

# A Notations and model description

In this appendix we first remind all the notations and settings of the attention-indexed models. We then remind mathematical concepts and definitions that are present in the main text.

Throughout this work, we use $\ell, k = 1, \ldots, L$ as the layer index where $L$ is the total number of layer matrices used, while $a, b = 1, \ldots, T$ are the token index and $T$ is the total number of tokens. Then $i, j = 1, \ldots, d$ are the indices for the dimensions and $d$ is the embedding dimension of each token, and $\mu = 1 \ldots n$ is the sample index and $n$ is the total number of samples. We will also use $u, v = 0 \ldots m$ as the replica indices from 0 to m.

We list below the specifics of our model:

- $X \equiv X_0 \in \mathbb{R}^{T \times d}$: The matrix of $T$ tokens (rows), each token of embedding dimension $d$.
- $S_\ell \in \mathbb{R}^{d \times d}$ symmetric matrices for $\ell = 1, \ldots, L$ and extracted independently from a rotationally invariant ensemble $P_S(S) = P_S(O^\top S O)$ for any rotation matrix $O$. We fix the normalizations such that $\mathbb{E}_{P_S}[\mathrm{Tr}\, S] = \kappa_1 d$ and $\mathbb{E}_{P_S}[\mathrm{Tr}\, S^2] = \kappa_2 d$ and with $\kappa_1, \kappa_2 = \mathcal{O}(1)$. Contextually, we assume that the empirical spectral distribution of $S$ will converge to a well defined measure $\mu_S$. For the purpose of the analysis, we will specify our general framework to symmetric matrices of the form $S_\ell = W^\top W / \sqrt{r_\ell d}$ where $W \in \mathbb{R}^{d \times r}$ with entries $W_{ij} \sim \mathcal{N}(0, 1)$. We refer to the finite quantities $\rho_l > 0$ as the width ratios of each layer.

- We define the AIM as the following model:

$$y = g\left(\{h^{(\ell)}\}_{\ell=1}^L\right) \tag{35}$$

with the generic map $g : \mathbb{R}^{L \times T \times T} \to \mathbb{R}^{T \times T}$ which depends on the quadratic preactivations

$$h_{ab}^{(\ell)} \equiv \frac{\boldsymbol{x}_a^\top S_\ell \, \boldsymbol{x}_b - \delta_{ab} \, \mathrm{Tr}\, S_\ell}{\sqrt{d}} \tag{36}$$

In the following appendix, we will show the tight link between the generic definition of the AIM with deep attention networks.

In the rest of this appendix, we recall the definition of the semicircle and Marchenko-Pastur laws in the context of random matrix theory. In particular

$$\sigma_{\mathrm{sc}, \Delta} = \frac{\sqrt{4\Delta - x^2}}{2\pi\Delta}\mathbb{I}\{|x| \leq 2\sqrt{\Delta}\},$$

$$\mu_{\mathrm{MP}, \rho}(x) = \begin{cases} (1-\rho)\delta(x) + \rho\frac{\sqrt{(\lambda_+ - x)(x - \lambda_-)}}{2\pi x}, & \text{if } \rho \leq 1 \\ \rho\frac{\sqrt{(\lambda_+ - x)(x - \lambda_-)}}{2\pi x}, & \text{if } \rho > 1 \end{cases} \tag{37}$$

Finally, we recall the following following definitions.

- Standard normal pdf and cdf

$$\phi(z) = \frac{e^{-z^2/2}}{\sqrt{2\pi}}, \qquad \Phi(z) = \int_{-\infty}^z \phi(t)\, dt = \tfrac{1}{2}\left(1 + \mathrm{erf}(z/\sqrt{2})\right) \tag{38}$$

- Bivariate normal density and cdf with correlation $c$

$$\phi_2(u, v; c) = \frac{\exp\left[-\frac{u^2 - 2cuv + v^2}{2(1-c^2)}\right]}{2\pi\sqrt{1-c^2}}, \quad \Phi_2(u, v; c) = \int_{-\infty}^u \int_{-\infty}^v \phi_2(t_1, t_2; c)\, dt_2\, dt_1. \tag{39}$$

We also remark that we formally define the Dirac delta function $\delta(x) = \lim_{\sigma \to 0} \mathcal{N}(0, \sigma)(x)$ as the limit to zero variance of a centered Gaussian.

We finally define the row-wise softmax function with inverse temperature $\beta$ acting on the matrix $h \in \mathbb{R}^{T \times T}$ matrix:

$$\sigma_\beta(h_{ab}) = \mathrm{Softmax}(\beta h_{ab}) = \frac{\exp(\beta h_{ab})}{\sum_b \exp(\beta h_{ab})} \tag{40}$$

# B    From deep self–attention to the *attention-indexed models*

In this appendix we highlight the connection between the AIM models defined in Eq. (1) with those of two crucial architectures employed in the analysis of Large Language Models (LLMs), namely deep attention networks and their sequence-to-sequence (seq2seq) version. In particular, we show that both the deep self–attention encoder and its sequence–to–sequence (seq2seq) variant can be rewritten exactly as an attention-indexed model of the form (1).

We keep the notation of the main text and the previous appendix: tokens are indexed by $a, b \in [T]$, embeddings by $\boldsymbol{x}_a^\top \in \mathbb{R}^d$, and every layer $l \in [L]$ carries a tied key–query weight matrix[1] $S_l \in \mathbb{R}^{d \times d}$ with extensive width $r_l = \rho_l d$ and rotationally–invariant prior $P_S$.

**Deep encoder.** Let $X_0 \in \mathbb{R}^{T \times d}$ be the matrix whose rows are the token embeddings, $(X_0)_{a:} = \boldsymbol{x}_a^\top$. A deep self–attention network with a residual (skip) connection and readout strength $c \geq 0$ is given by the recursive formula:

$$X_\ell \;=\; \Big[ c\,\mathbb{I}_T + \sigma_\beta\big(H_\ell(X_{\ell-1})\big) \Big] X_{\ell-1}, \qquad \ell = 1, \ldots, L, \tag{41}$$

where $\sigma : \mathbb{R}^{T \times T} \to \mathbb{R}^{T \times T}$ is the row–wise softmax with inverse temperature $\beta > 0$ implicitly contained in the symbol $\sigma(\cdot)$. The function $H_{\ell-1}(X_{\ell-1})$ is given by:

$$H_\ell(X_{\ell-1}) \;=\; \frac{1}{\sqrt{d}} X_{\ell-1} S_\ell X_{\ell-1}^\top - \frac{1}{\sqrt{d}} \mathbb{E}_{\mathrm{tr}}[X_{\ell-1} S_\ell X_{\ell-1}^\top] \tag{42}$$

where the expectation is intended over the input data $X_0$. Define the preactivations

$$h_{ab}^{(\ell)} \;:=\; \frac{1}{\sqrt{d}} \boldsymbol{x}_a^\top S_\ell \boldsymbol{x}_b - \frac{1}{\sqrt{d}} \operatorname{Tr} S_\ell\, \delta_{ab}, \qquad \ell = 1, \ldots, L,\; a, b \in [T], \tag{43}$$

and the sequence of token–space operators

$$B_c^0 := \mathbb{I}_T, \qquad B_c^\ell := \Big[ c\,\mathbb{I}_T + \sigma_\beta\big( B_c^{\ell-1} h^{(\ell)} B_c^{\ell-1\top} \big) \Big] B_c^{\ell-1}, \quad \ell = 1, \ldots, L, \tag{44}$$

One verifies inductively that

$$X_\ell = B_c^\ell X_0, \qquad \ell = 0, \ldots, L, \tag{45}$$

so that every hidden representation depends on the data only through the collection $\big\{ h^{(1)}, \ldots, h^{(\ell)} \big\}$. In this way, we can write:

$$H_\ell(X_{\ell-1}) = \frac{1}{\sqrt{d}} X_{\ell-1} S_\ell X_{\ell-1}^\top - \frac{1}{\sqrt{d}} \mathbb{E}_{\mathrm{tr}}[X_{\ell-1} S_\ell X_{\ell-1}^\top] \tag{46}$$

$$= \frac{1}{\sqrt{d}} X_{\ell-1} S_\ell X_{\ell-1}^\top - \frac{1}{\sqrt{d}} \mathbb{E}_{\mathrm{tr}}[B_c^{\ell-1} X_0 S_\ell X_0^\top B_c^{\ell-1\top}] \tag{47}$$

$$= \frac{1}{\sqrt{d}} X_{\ell-1} S_\ell X_{\ell-1}^\top - \frac{\operatorname{Tr}S_\ell}{\sqrt{d}} B_c^{\ell-1} B_c^{\ell-1\top} \tag{48}$$

Which is exactly the formula shown in (4). Furthermore, this map does only depend on the preactivations $h^{(\ell)}$ in the following way:

$$H_\ell(X_{\ell-1}) = \frac{1}{\sqrt{d}} (B_c^{\ell-1} X_0) S_\ell (B_c^{\ell-1} X_0)^\top - \frac{\operatorname{Tr}S_\ell}{\sqrt{d}} B_c^{\ell-1} B_c^{\ell-1\top} \tag{49}$$

$$= B_c^{\ell-1} \Big( \frac{1}{\sqrt{d}} X_0 S_\ell X_0^\top \Big) B_c^{\ell-1\top} - \frac{\operatorname{Tr}S_\ell}{\sqrt{d}} B_c^{\ell-1} B_c^{\ell-1\top} \tag{50}$$

$$= B_c^{\ell-1} \Big( h^{(\ell)} + \frac{\operatorname{Tr}S_\ell}{\sqrt{d}} \mathbb{I}_T \Big) B_c^{\ell-1\top} - \frac{\operatorname{Tr}S_\ell}{\sqrt{d}} B_c^{\ell-1} B_c^{\ell-1\top} \tag{51}$$

$$= B_c^{\ell-1} h^{(\ell)} B_c^{\ell-1\top} \tag{52}$$

---

[1] For simplicity we restrict to the single–head, tied setting; extending to multi–head merely introduces an additional block index.

where we used the definition in (1). This completes our mapping in (7). In particular the *deep-attention output* is given by:

$$y = \sigma_\beta\Big(H_L(X_{L-1})\Big) = g_{\text{deep}}\Big(h^{(1)}, \ldots, h^{(L)}\Big) \in \mathbb{R}^{T \times T}, \tag{53}$$

with[2] $g_{\text{deep}}(h^{(1)}, \ldots, h^{(L)}) := \sigma_\beta\big(B_c^{L-1}(h^{(1:L-1)})\, h^{(L)}\, B_c^{L-1}(h^{(1:L-1)})^\top\big)$. Equation (53) is *exact* and has the attention-indexed model structure (1): the whole deep network collapses to a deterministic multivariate function $g_{\text{deep}}$ of the $L$ bilinear indices $h_{ab}^{(\ell)} \sim \{\boldsymbol{x}_a^\top S_\ell \boldsymbol{x}_b\}_{\ell,a,b}$.

**Seq2seq variant.** If the last layer keeps the token embeddings instead of collapsing them, i.e.

$$X_L = \sigma_\beta\Big(H_L(X_{L-1})\Big) X_{L-1}, \tag{54}$$

with exactly the same algebra

$$X_L = g_{\text{seq}}\Big(h^{(1)}, \ldots, h^{(L)}\Big) X_0, \qquad g_{\text{seq}}(h^{(1:L)}) := \sigma_\beta\big(B_c^{L-1} h^{(L)} B_c^{L-1\top}\big) B_c^{L-1}. \tag{55}$$

Thus the seq2seq readout is also an attention-indexed model: a (matrix-valued) function of the same quadratic statistics, followed by a fixed linear map $X_0$.

Note that in the particular case of just $L = 1$ layer the seq2seq map simplifies into:

$$X_1 = g_{\text{seq}}\Big(\{h^{(1)}\}_{a \leq b}^T\Big) X_0, \quad g_{\text{seq}}(h^{(1)}) = \sigma_\beta\big(B_c^0 h^{(1)} B_c^{0\top}\big) B_c^0 = \sigma_\beta(h^{(1)}) = g_{deep}(h^{(1)}) \tag{56}$$

From this paragraph we can hence conclude that, as shown in equations (53) and (55), any $L$-layer tied self-attention network with extensive-width weights is information-theoretically equivalent to an attention-indexed model with $L$ indices. Consequently all the Bayes–optimal analysis carried out in Secs. 2–4 applies verbatim to deep self-attention and to its seq2seq counterpart: learning the matrices $\{S_\ell\}$ under the deep architecture is statistically equivalent to learning them under the attention-indexed model (1).

**Multi-head self-attention.** Let heads $m = 1, \ldots, M$ with (tied) weights $S_\ell^{(m)}$ and per-head logits $H_\ell^{(m)}(X_{\ell-1}) = \frac{1}{\sqrt{d}} X_{\ell-1} S_\ell^{(m)} X_{\ell-1}^\top - \mathbb{E}_{\text{tr}}[\frac{1}{\sqrt{d}} X_{\ell-1} S_\ell^{(m)} X_{\ell-1}^\top]$. A standard multi-head layer computes head-wise weights $\sigma_\beta(H_\ell^{(m)})$ and then aggregates (by concatenation or averaging) before a token-wise linear map; for our purposes, averaging:

$$X_\ell = \Big[c\,\mathbb{I}_T + \frac{1}{M} \sum_{m=1}^{M} \sigma_\beta\big(H_\ell^{(m)}(X_{\ell-1})\big)\Big] X_{\ell-1}. \tag{57}$$

Applying the transport identity headwise and averaging gives

$$\frac{1}{M} \sum_{m=1}^{M} H_\ell^{(m)}(X_{\ell-1}) = B_c^{\ell-1}\Big(\frac{1}{M} \sum_{m=1}^{M} h^{(\ell,m)}\Big) B_c^{\ell-1\top}. \tag{58}$$

If we define the quantity $\bar{h}^{(\ell)} = \frac{1}{M} \sum_{m=1}^{M} h^{(\ell,m)}$, hence, the closed recursion is

$$B_c^\ell = \Big[c\,\mathbb{I}_T + \sigma_\beta\Big(B_c^{\ell-1}\, \bar{h}^{(\ell)}\, B_c^{\ell-1\top}\Big)\Big] B_c^{\ell-1}, \quad B_c^0 = \mathbb{I}_T, \tag{59}$$

with output

$$y = \sigma_\beta\Big(B_c^{L-1}\, \bar{h}^{(L)}\, B_c^{L-1\top}\Big), \tag{60}$$

and the seq2seq variant

$$X_L = \sigma_\beta\Big(B_c^{L-1}\, \bar{h}^{(L)}\, B_c^{L-1\top}\Big) B_c^{L-1} X_0. \tag{61}$$

Finally, also this multi-head variant of the model is again an AIM. Here, the collection of indices is simply enlarged to $\{h^{(\ell,m)}\}_{\ell=1..L,\, m=1..M}$, with each $h^{(\ell,m)}$ defined as in (1). The extensive-rank regime corresponds to ranks $r_\ell^{(m)} = \rho_\ell^{(m)} d$.

---

[2]The explicit form of $g_{\text{deep}}$ is obtained by inserting (45) with $l = L - 1$.

## C  Bayes optimal analysis of attention-indexed models (AIM)

We study a model described by the general setting:

$$y_\mu \sim P_{\text{out}}\Big(\frac{\boldsymbol{x}_a^{\mu\top} S_\ell\, \boldsymbol{x}_b^\mu - \delta_{ab}\operatorname{Tr} S_\ell}{\sqrt{d}}\Big)_{\ell=1,\ldots,L}^{a,b=1,\ldots,T} \tag{62}$$

with $\boldsymbol{x}_a^\top$ rows of $X \in \mathbb{R}^{T\times d}$, $S_\ell \in \mathbb{R}^{d\times d}$ symmetric and $y \in \mathbb{R}^{T\times T}$. Indices range from $\mu = 1\ldots n$ samples, with $d, n >> 1$. Instead the number of tokens and layers $T, L << d$: we interpret Eq. (62) as $y_\mu$ outputs generated by a model of attention from data $X$ that are processed in a bilinear way through:

$$y_\mu = g\left(\{\frac{\boldsymbol{x}_a^\mu S_\ell\, \boldsymbol{x}_b^{\mu\top} - \delta_{ab}\operatorname{Tr} S_\ell}{\sqrt{d}}\}_{\ell=1}^L\right). \tag{63}$$

or following Eq. (6):

$$y_\mu = g_{\text{deep}}(h^{(1)}, \ldots, h^{(L)}) = B_c^L\left(\{\frac{\boldsymbol{x}_a^{\mu\top} S_\ell\, \boldsymbol{x}_b^\mu - \delta_{ab}\operatorname{Tr} S_\ell}{\sqrt{d}}\}_{a,b=1\ldots T}^{\ell=1\ldots L}\right) \in \mathbb{R}^{T\times T} \tag{64}$$

and

$$P_{\text{out}}\Big(\frac{\boldsymbol{x}_a^{\mu\top} S_\ell\, \boldsymbol{x}_b^\mu - \delta_{ab}\operatorname{Tr} S_\ell}{\sqrt{d}}\Big)_{\ell=1,\ldots,L}^{a,b=1,\ldots,T} = \delta\Big(y - B_c^L\big(\frac{\boldsymbol{x}_a^{\mu\top} S_\ell\, \boldsymbol{x}_b^\mu - \delta_{ab}\operatorname{Tr} S_\ell}{\sqrt{d}}\big)\Big) \tag{65}$$

In our setting, the matrices $S_\ell$ are symmetrical for each layer $\ell$ and we consider multiple layers indices $\ell = 1, \ldots, L$. $\boldsymbol{x}_a^\top$ is the a-th row of $X$ for $a, b = 1, \ldots, T$. Each row $\boldsymbol{x}_a^\top \in \mathbb{R}^d$ has i.i.d. Gaussian entries, so $x_{ai}^\mu \sim \mathcal{N}(0,1)$.

We define the preactivations

$$h_{ab}^{(\ell)\,\mu} = \frac{\boldsymbol{x}_a^{\mu\top} S_\ell\, \boldsymbol{x}_b^\mu - \delta_{ab}\operatorname{Tr} S_\ell}{\sqrt{d}} \tag{66}$$

Since the matrices $S_\ell$ are symmetric, so are the preactivations of the model. Finally, for convenience, we rewrite the preactivations of the model in terms of the symmetrized sensing matrices

$$Z_{ij,ab}^\mu \equiv (x_{i,a}^\mu x_{j,b}^\mu + x_{j,a}^\mu x_{i,b}^\mu - 2\delta_{ij}\delta_{ab})/\sqrt{2d(1+\delta_{ab})} \in \mathbb{R} \tag{67}$$

The preactivations of the model can thus be expressed as:

$$\sqrt{2-\delta_{ab}}\, h_{ab}^{(\ell)\,\mu} = \operatorname{Tr}(S_\ell Z_{ab}^\mu) = \tilde{h}_{ab}^{(\ell)\,\mu} \tag{68}$$

In the rest of the analysis, we will refer to this equivalent representation of the model by considering symmetrized data $\tilde{h}_{ab}^{(\ell)\,\mu}$ that we will just recall $h_{ab}^{(\ell)\,\mu}$, while incorporating the factor $\sqrt{2-\delta_{ab}}$ in the output function part.

### C.1  Replica analysis of AIM and their state evolution

Starting from the posterior distribution of the model:

$$P(S_1, \ldots, S_L|\mathcal{D}) = \frac{1}{\mathcal{Z}(\mathcal{D})} \prod_{\ell=1}^L P_S(S_\ell) \prod_{\mu=1}^n \delta\Big(y^\mu - g\big(h^{(1)}(S_1, \boldsymbol{x}^\mu), \ldots, h^{(L)}(S_L, \boldsymbol{x}^\mu)\big)\Big), \tag{69}$$

the replicated partition function of the model in Eq. (62) is:

$$\langle \mathcal{Z}(\mathcal{D})^m \rangle = \mathbb{E}_{y,X} \int \prod_{\ell=1}^L \prod_{u=0}^m dS_\ell^u\, P_0(S_\ell^u) \prod_{\mu=1}^n \prod_{a\leq b}^T P_{\text{out}}\Big(y^\mu \,\Big|\, \Big\{\frac{h_{ab}^{(\ell),\mu,u}}{\sqrt{2-\delta_{ab}}}\Big\}_{ab}\Big) \delta\Big(h_{ab}^{(\ell),\mu,u} - \operatorname{Tr}(S_\ell^u Z_{ab}^\mu)\Big) \tag{70}$$

where $P_0(S_\ell^u)$ is the rotational invariant prior distribution of each $S_\ell$, and $h_{ab}^{(\ell),\mu,u}$ are the replicated preactivations in terms of the symmetrized data as explained in (68). $u$ is the replica index, we work in a Bayes optimal setting. Above, $\mu \in \{1, \ldots, n\}$ enumerates data samples, $\ell \in \{1, \ldots, L\}$ indexes the distinct layers, $u \in \{0, \ldots, m\}$ indexes the replicas, and $a, b \in \{1, \ldots, T\}$ are the token indices.

We compute the expectation with respect to the data exploiting the Gaussian-equivalence principle:

$$\mathbb{E}_X \delta\Big(h_{ab}^{(\ell),\mu,u} - \mathrm{Tr}(S_\ell^u Z_{ab}^\mu)\Big) \quad \mapsto \quad P_h\Big(\{h_{ab}^{(\ell),\mu,u}\}_{\ell,\mu,a,b,u}\Big), \tag{71}$$

where $P_h$ is a joint Gaussian distribution with the means and covariances:

$$\mathbb{E}\big[h_{ab}^{(\ell),\mu,u} - \mathrm{Tr}(S_\ell^u Z_{ab}^\mu)\big] = 0, \quad \mathrm{Cov}_{x^\mu}\Big(h_{a\leq b}^{(\ell)\,u}, h_{c\leq d}^{(k)\,v}\Big) = \frac{1}{d}\,[2\,\delta_{ac}\delta_{bd}]\,\mathrm{Tr}\,(S_\ell^u S_k^v) \tag{72}$$

We introduce the order parameters measuring the $S_\ell^u$–$S_k^v$ overlaps:

$$Q_{\ell k}^{uv} := \frac{1}{d}\,\mathrm{Tr}\Big(S_\ell^u\,S_k^v\Big), \quad \text{for } \ell, k = 1, \ldots, L,\ u, v = 0, \ldots, m. \tag{73}$$

We enforce the definitions of the overlaps by inserting $\delta$-functions:

$$\prod_{\ell,k=1}^{L} \prod_{u\leq v=0}^{m} \delta\Big(d^2\,Q_{\ell k}^{uv} - d\,\mathrm{Tr}\big[S_\ell^u\,S_k^v\big]\Big), \tag{74}$$

and introduce the corresponding conjugate fields $\widehat{Q}_{\ell k}^{uv}$. We insert

$$\delta\Big(d^2\,Q_{\ell k}^{uv} - d\,\mathrm{Tr}[S_\ell^u\,S_k^v]\Big) = \int \mathrm{d}\widehat{Q}_{\ell k}^{uv}\,\exp\Big\{ i\,\frac{\widehat{Q}_{\ell k}^{uv}}{2}\Big(d^2\,Q_{\ell k}^{uv} - d\,\mathrm{Tr}[S_\ell^u\,S_k^v]\Big)\Big\}. \tag{75}$$

Hence the replicated partition function can be schematically written:

$$\langle \mathcal{Z}(\mathcal{D})^m \rangle = \int \Big(\prod_{u,\ell} \mathrm{d}S_\ell^u\,P_0(S_\ell^u)\Big) \int \Big(\prod_{u\leq v,\ell,k} \mathrm{d}Q_{\ell k}^{uv}\,\mathrm{d}\hat{Q}_{\ell k}^{uv}\Big)$$

$$\times \exp\Big[\frac{i}{2}\sum_{u\leq v,\ell,k} \hat{Q}_{\ell k}^{uv}\big(d^2\,Q_{\ell k}^{uv}\big)\Big] \times \exp\Big[-\frac{i\,d}{2}\sum_{u\leq v,\ell,k} \hat{Q}_{\ell k}^{uv}\,\mathrm{Tr}(S_\ell^u S_k^v)\Big]$$

$$\times \prod_{\mu=1}^{n}\Big[\int\prod_{u,\ell} \mathrm{d}h_{ab}^{(\ell),\mu,u}\,P_h\big(h^{(\ell),\mu,u}\big) \prod_{u,\ell,a\leq b} P_{\mathrm{out}}\big(y_{ab}^\mu \mid \{\frac{h_{ab}^{(\ell),\mu,u}}{\sqrt{2-\delta_{ab}}}\}_{ab}\big)\Big], \tag{76}$$

In a replica-symmetric (RS) scenario, we let

$$Q_{\ell k}^{uv} = \begin{cases} Q_{\ell k}, & (u = v), \\ q_{\ell k}, & (u \neq v). \end{cases} \tag{77}$$

and:

$$i\hat{Q}_{\ell k}^{uv} = \begin{cases} \hat{Q}_{\ell k}, & \text{if } u = v \\ -\hat{q}_{\ell k}, & \text{if } u \neq v \end{cases} \tag{78}$$

Hence, e.g. the exponent $\sum_{\ell,k,u,v} i\,\widehat{Q}_{\ell,k}^{uv}\,d^2\,Q_{\ell,k}^{uv}$ becomes

$$i\,d^2 \sum_{\ell,k}\Big[\frac{(m+1)}{2}\,\widehat{Q}_{\ell k}\,Q_{\ell k} - \frac{m(m+1)}{4}\,\widehat{q}_{\ell k}\,q_{\ell k}\Big]. \tag{79}$$

Likewise, $-\sum_{\ell,k,u,v} \widehat{Q}_{\ell k}^{uv}\mathrm{Tr}(S_\ell^u S_k^v)$ can be reorganized in a form that leads in the limit $m \to 0$ to typical terms $\widehat{Q}_{\ell k}^{uu} = 0$ or similar. Moreover $\widehat{Q}_{\ell k}^{uv} = -\frac{\hat{q}_{\ell k}}{2}$.

So finally the replicated partition function, hence, takes the following form:

$$\langle \mathcal{Z}(\mathcal{D})^m \rangle = \int \prod_{u\leq v,\ell,k} dQ_{\ell k}^{uv} d\hat{Q}_{\ell k}^{uv}\,\exp\Big(\frac{i}{2}d^2 \sum_{u\leq v,\ell,k} \hat{Q}_{\ell k}^{uv} Q_{\ell k}^{uv}\Big) I_{\mathrm{in}} I_{\mathrm{out}} \tag{80}$$

with:

$$d^2 I_{\mathrm{in}}(\hat{q}) = \int \prod_{u,\ell} dS_\ell^u\,P_0(S_\ell^u)\,\exp\Big(-\frac{i\,d}{2}\sum_{u\leq v,\ell,k} \hat{Q}_{\ell k}^{uv}\,\mathrm{Tr}(S_\ell^u S_k^v)\Big), \tag{81}$$

$$I_{\mathrm{out}}(q) = \Big[\int dy \int \prod_{u,\ell,a\leq b} dh_{ab}^{(\ell)\,u}\,P\big(h_{ab}^{(\ell)\,u}\big) \prod_{u,\ell} P_{\mathrm{out}}\Big(y\mid\{\frac{h_{ab}^{(\ell)\,u}}{\sqrt{2-\delta_{ab}}}\}_{ab}\Big)\Big]^n. \tag{82}$$

The free entropy per degree of freedom of the problem is defined as

$$\Phi = \lim_{d\to\infty} \frac{1}{d^2} \lim_{n\to\infty} \lim_{m\to 0} \frac{1}{m} \ln\langle Z^m\rangle .\tag{83}$$

After introducing $n = \alpha d^2$ data samples, the free entropy decomposes into a prior contribution and an output contribution:

$$\Phi = \text{extr}_{\{q,\hat{q}\}} \left\{ -\frac{\text{Tr}\,q\hat{q}}{4} + I_{\text{in}}(\hat{q}) + \alpha I_{\text{out}}(q) \right\} .\tag{84}$$

Thus obtaining the state equations:

$$q = 4\,\partial_{\hat{q}} I_{\text{in}}(\hat{q})\tag{85}$$
$$\hat{q} = 4\alpha\,\partial_q I_{\text{out}}(q)\tag{86}$$

## C.2 Prior Term Computation

First we compute under the RS ansatz:

$$-\frac{i\,d}{2} \sum_{u\leq v=0}^{m} \hat{Q}_{\ell k}^{uv} \,\text{Tr}\,(S_\ell^u S_k^v) = -\frac{i\,d}{2}\left( \sum_{u=0}^{m} \hat{Q}_{\ell k}\,\text{Tr}\,(S_\ell^u S_k^u) + \sum_{u<v}(-\hat{q}_{\ell k})\,\text{Tr}\,(S_\ell^u S^v) \right)\tag{87}$$

$$= -\frac{\hat{Q}_{\ell k}\,d}{2} \sum_{u=0}^{m} \text{Tr}\,(S_\ell^u S_k^u) + \frac{\hat{q}_{\ell k}\,d}{2} \sum_{u<v} \text{Tr}\,(S_\ell^u S_k^v)\tag{88}$$

$$= -\frac{d}{2}\left(\hat{Q}_{\ell k} + \frac{\hat{q}_{\ell k}}{2}\right) \sum_{u=0}^{m} \text{Tr}\,(S_\ell^u S_k^u) + \frac{\hat{q}_{\ell k}\,d}{4} \sum_{u,v=0}^{m} \text{Tr}\,(S_\ell^u S_k^v)\tag{89}$$

We remind that each $S_\ell$ is a rank-$\rho_\ell d$ rotationally invariant matrix of order $O(d \times d)$. The prior factor that emerges from the partition function, after decoupling the replica indices by applying a Hubbard-Stratonovich transformation, reads:

$$I_{\text{in}}(\hat{q}) = \int \prod_{\ell=1}^{L} \prod_{u=0}^{m} dS_\ell^u\, P_0(S_\ell^u) \exp\left\{ -\frac{i\,d}{2} \sum_{\ell,k=1}^{L} \sum_{u\leq v=0}^{m} \widehat{Q}_{\ell k}^{(u,v)}\,\text{Tr}\big(S_\ell^u S_k^v\big) \right\}\tag{90}$$

$$= \int d\bar{S}P_0(\bar{S}) \exp\left\{ \sum_{\ell,k} -\frac{d}{2}\left(\hat{Q}_{\ell k} + \frac{\hat{q}_{\ell k}}{2}\right) \sum_{u=0}^{m} \text{Tr}\,(S_\ell^u S_k^u) + \frac{\hat{q}_{\ell k}\,d}{4} \sum_{u,v=0}^{m} \text{Tr}\,(S_\ell^u S_k^v) \right\}\tag{91}$$

$$= \int d\bar{S}P_0(\bar{S}) \exp\left\{ -\sum_{\ell,k}\sum_{u} \frac{\hat{q}_{\ell,k}\,d}{4} \text{Tr}(S_\ell^u S_k^u) + \sum_{\ell k}\sum_{u,v} \frac{\hat{q}_{\ell k}d}{4} \text{Tr}(S_\ell^u S_k^v) \right\}\tag{92}$$

$$= \int d\bar{S}P_0(\bar{S})\mathcal{D}(Y) \exp\left\{ -\sum_{\ell,k}\sum_{u} \frac{\hat{q}_{\ell k}\,d}{4} \text{Tr}(S_\ell^u S_k^v) + \sum_{\ell,k}\sum_{u} \frac{\sqrt{\hat{q}_{\ell k}}\,d}{2} \text{Tr}(S_k^u Y_\ell) \right\}\tag{93}$$

$$= \int \mathcal{D}(Y)\left\{ \int d\bar{S}P_0(\bar{S}) \exp\left\{ -\frac{d}{4} \sum_{\ell,k}^{L} \hat{q}_{\ell k}\,\text{Tr}(S_\ell S_k) + d\sum_{\ell,k}^{L} \frac{\sqrt{\hat{q}_{\ell k}}}{2}\,\text{Tr}(S_k Y_\ell) \right\} \right\}^{m+1}\tag{94}$$

where $\mathcal{D}(Y_\ell)$ are GOE(d) measures $\forall \ell \in [L]$ and $Y_\ell \in \mathbb{R}^{d\times d}$ and also $\bar{S} \in [\mathbb{R}^{d\times d}]^L$. In Eq.(78) we used the identity:

$$\mathbb{E}_{Y\sim\text{GOE}(d)}\left[ e^{\frac{d}{2}\,\text{Tr}[SY]} \right] = e^{\frac{d}{4}\,\text{Tr}[S^2]}$$

Finally, taking the zero replica $m \to 0$ limit, we can write the prior contribution to the free entropy of the model as:

$$I_{\text{in}}(\hat{q}) = \lim_{d\to\infty} \frac{1}{d^2} \int DY_1 \ldots DY_L\, \mathcal{Z}_{\text{in}}(Y_1, \ldots, Y_L; \hat{q}) \log \mathcal{Z}_{\text{in}}(Y_1, \ldots, Y_L; \hat{q})$$

$$\mathcal{Z}_{\text{in}}(\{Y_\ell\}_{\ell=1}^{L}; \hat{q}) = \int \left[ \prod_{\ell=1}^{L} dS_\ell\, P_S(S_\ell) \right]$$

$$\times \exp\left[ -\frac{d}{4} \sum_{\ell,k=1}^{L} \hat{q}_{\ell k}\,\text{Tr}(S_\ell S_k) + \frac{d}{2} \sum_{\ell,k=1}^{L} \sqrt{\hat{q}}_{\ell k}\,\text{Tr}(Y_\ell S_k) \right].\tag{95}$$

The matrices $Y_\ell \in \mathbb{R}^{d \times d}$ are the auxiliary fields introduced by the Hubbard–Stratonovich transformation. Notably, they can be interpreted as "noisy measurements" of the $S_\ell$ matrices with coupled indices. In particular, the denoising problem which is solved by the free-entropy contribution of the prior is:

$$Y_\ell^{ij} = \sum_k \sqrt{\hat{q}_{\ell k}} \, S_k^{ij} + Z_\ell^{ij} \quad \forall i, j, \ell \tag{96}$$

with $Z_\ell$ GOE(d) matrices and $S_\ell \in \mathbb{R}^{d \times d}$ rotationally invariant matrices, leading to an exponential term of the form of $-\frac{1}{2} \sum_\ell \mathrm{Tr}((\sum_k \sqrt{\hat{q}_{\ell k}} S_k - Y_\ell)^2)$. Such equivalence between the matrix denoising problem in (96) and (95) is analogous to those of [42, 40].

## C.3 Output Channel Computation

Starting from the replicated partition function in Eq. (80), we can see that the output channel contribution to the free entropy of the model, factorized with respect to the data, is given by:

$$I_{\text{out}}(q) = \left[ \int dy \int \prod_{u,\ell,a \leq b} dh_{ab}^{(\ell)\,u} \, P\left(\{h_{ab}^{(\ell)\,u}\}_{ab}\right) \prod_{u,\ell} P_{\text{out}}\left(y \mid \left\{\frac{h_{ab}^{(\ell)\,u}}{\sqrt{2 - \delta_{ab}}}\right\}_{ab}\right) \right]^n, \tag{97}$$

where we consider only the upper triangular token indices $a \leq b$.
$P_h\left(\{h_{ab}^{(\ell),u}\}_{ab}\right)$ is a multivariate Gaussian distribution with means and covariance:

$$\mathbb{E}[h_{ab}^{(\ell)}] = 0, \qquad \mathrm{Cov}_{x^\mu}\left(h_{a \leq b}^{(\ell)\,u}, h_{c \leq d}^{(k)\,v}\right) = \frac{1}{d}\left[2\,\delta_{ac}\delta_{bd}\right] \mathrm{Tr}\left(S_\ell^u S_k^v\right) = \left[2\,\delta_{ac}\delta_{bd}\right] Q_{\ell k}^{uv}. \tag{98}$$

Under the RS ansatz and in the limit $m \to 0$, we can decouple the replicas through another Hubbard-Stratonovich transformation. The exponent involving $h_{ab}^{(\ell)\,u}$ becomes:

$$-\frac{1}{2} \sum_{u,v=0}^m \sum_{a \leq b, c \leq d} \sum_{\ell,k} \left(h_{ab}^{(\ell)\,u}\right) \left(\Sigma_h^{-1}\right)_{ab,cd}^{uv,\ell k} \left(h_{cd}^{(k)\,v}\right). \tag{99}$$

Substituting back, the output term becomes:

$$I_{\text{out}}(q) = \left[ \int dy \int \prod_{u,\ell,a \leq b} dh_{ab}^{(\ell)\,u} \exp\left(-\frac{1}{2} \sum_{u,v=0}^m \sum_{a,b,c,d} \sum_{\ell,k} h_{ab}^{(\ell)\,u} \left(\Sigma_h^{-1}\right)_{a \leq b, c \leq d}^{uv,\ell k} h_{cd}^{(k)\,v}\right) \prod_{u,\ell} P_{\text{out}}\left(y \mid \left\{\frac{h_{ab}^{(\ell)\,u}}{\sqrt{2 - \delta_{ab}}}\right\}_{ab}\right) \right]^n. \tag{100}$$

For a fixed channel $\ell$ and for each token pair $(a, b)$ with $a \leq b$, the covariance in the replica space is given by

$$(\Sigma_h)_{a \leq b, c \leq d}^{uv,\ell k} = \left[2\,\delta_{ac}\delta_{bd}\right] Q_{\ell k}^{uv} = \left[2\,\delta_{ac}\delta_{bd}\right] \left[(Q_{\ell k} - q_{\ell k})\,\delta_{uv} + q_{\ell k}\right]. \tag{101}$$

Because of the Kronecker structure $\delta_{ac}\delta_{bd}$, the Gaussian law over all $\{h_{ab}^{(\ell)u}\}_{a \leq b, \ell, u}$ factorizes over token pairs $(a, b)$. Hence it is sufficient to treat one fixed pair $(a, b)$ and then take the product over $a \leq b$. For notational clarity in the next steps, we temporarily fix a pair $(a, b)$ and write $h^{(\ell)u} \equiv h_{ab}^{(\ell)u}$. For this pair, the covariance across replicas and layers reads

$$\mathbb{E}\left[h^{(\ell)u} h^{(k)v}\right] = 2\left[(Q_{\ell k} - q_{\ell k})\,\delta_{uv} + q_{\ell k}\right], \qquad u, v = 0, \ldots, m, \ \ \ell, k = 1, \ldots, L. \tag{102}$$

We now construct explicitly a family of Gaussian random variables with covariance (102). Introduce a shared Gaussian vector $\omega = (\omega^{(1)}, \ldots, \omega^{(L)})$ with mean zero and covariance

$$\mathbb{E}\left[\omega^{(\ell)} \omega^{(k)}\right] = 2q_{\ell k}. \tag{103}$$

For each replica $u = 0, \ldots, m$, an independent Gaussian vector $\xi^u = (\xi^{(1)u}, \ldots, \xi^{(L)u})$ with mean zero and covariance

$$\mathbb{E}\left[\xi^{(\ell)u} \xi^{(k)v}\right] = 2(Q_{\ell k} - q_{\ell k})\,\delta_{uv}. \tag{104}$$

$\omega$ is independent of all $\{\xi^u\}_{u=0}^m$. Define for each replica $u$ and layer $\ell$:

$$h^{(\ell)u} := \omega^{(\ell)} + \xi^{(\ell)u}. \tag{105}$$

Then, for all $\ell, k$ and $u, v$,

$$
\begin{aligned}
\mathbb{E}\Big[h^{(\ell)u}h^{(k)v}\Big] &= \mathbb{E}\Big[(\omega^{(\ell)} + \xi^{(\ell)u})(\omega^{(k)} + \xi^{(k)v})\Big] \\
&= \mathbb{E}\Big[\omega^{(\ell)}\omega^{(k)}\Big] + \mathbb{E}\Big[\xi^{(\ell)u}\xi^{(k)v}\Big] \\
&= 2q_{\ell k} + 2(Q_{\ell k} - q_{\ell k})\delta_{uv} = 2\Big[(Q_{\ell k} - q_{\ell k})\delta_{uv} + q_{\ell k}\Big], \quad (106)
\end{aligned}
$$

which is exactly (102). Therefore the law of $\{h^{(\ell)u}\}_{\ell,u}$ is exactly the RS Gaussian law.

From (105), conditional on $\omega$ the replicas are independent and

$$
\{h^{(\ell)u}\}_{\ell=1}^{L} \,\Big|\, \omega \;\sim\; \mathcal{N}\Big(\{\omega^{(\ell)}\}_{\ell=1}^{L},\, V\Big), \qquad V^{(\ell k)} := 2(Q_{\ell k} - q_{\ell k}). \quad (107)
$$

Equivalently, for each $u$,

$$
p\Big(\{h^{(\ell)u}\}_\ell \mid \omega\Big) = \frac{1}{\sqrt{\det(2\pi V)}}\, \exp\left(-\frac{1}{2}\sum_{\ell=1}^{L}\sum_{k=1}^{L}\big(h^{(\ell)u} - \omega^{(\ell)}\big)[V^{-1}]_{\ell k}\big(h^{(k)u} - \omega^{(k)}\big)\right). \quad (108)
$$

Moreover $\omega \sim \mathcal{N}(0, 2q)$, i.e.

$$
p(\omega) = \frac{1}{\sqrt{\det(2\pi\, 2q)}}\, \exp\left(-\frac{1}{2}\sum_{\ell=1}^{L}\sum_{k=1}^{L}\omega^{(\ell)}\,[(2q)^{-1}]_{\ell k}\,\omega^{(k)}\right). \quad (109)
$$

By marginalization over $\omega$, the joint density of $\{h^u\}_{u=0}^m$ is therefore:

$$
p\Big(\{h^{(\ell)u}\}_{\ell,u}\Big) = \int d^L\omega\, p(\omega)\prod_{u=0}^{m} p\Big(\{h^{(\ell)u}\}_\ell \mid \omega\Big). \quad (110)
$$

We now express $\omega \sim \mathcal{N}(0, 2q)$ through standard normals. Introduce i.i.d. auxiliary variables $\eta^{(1)}, \ldots, \eta^{(L)} \sim \mathcal{N}(0, 1)$ and define

$$
\omega^{(\ell)} = \sum_{r=1}^{L}(\sqrt{2q})_{\ell r}\, \eta^{(r)}, \quad (111)
$$

where $(\sqrt{2q})$ is any matrix square root such that, for all $\ell, k$,

$$
\sum_{r=1}^{L}(\sqrt{2q})_{\ell r}(\sqrt{2q})_{kr} = (2q)_{\ell k}. \quad (112)
$$

Then $\omega$ has covariance (103) since

$$
\mathbb{E}[\omega^{(\ell)}\omega^{(k)}] = \sum_{r,s}(\sqrt{2q})_{\ell r}(\sqrt{2q})_{ks}\,\mathbb{E}[\eta^{(r)}\eta^{(s)}] = \sum_{r}(\sqrt{2q})_{\ell r}(\sqrt{2q})_{kr} = (2q)_{\ell k}. \quad (113)
$$

Consequently, for any function $F(\omega)$,

$$
\int d^L\omega\, \mathcal{N}(\omega; 0, 2q)\, F(\omega) = \int \prod_{r=1}^{L}\frac{d\eta^{(r)}}{\sqrt{2\pi}}e^{-(\eta^{(r)})^2/2}\, F\left(\left\{\sum_{r}(\sqrt{2q})_{\ell r}\eta^{(r)}\right\}_{\ell=1}^{L}\right). \quad (114)
$$

Restoring the token indices, for each $(a, b)$ we introduce independent $\eta_{ab}^{(r)} \sim \mathcal{N}(0, 1)$ and define

$$
\omega_{ab}^{(\ell)} = \sum_{k=1}^{L}(\sqrt{2q})_{\ell k}\, \eta_{ab}^{(k)}, \qquad V^{(\ell k)} = 2(Q_{\ell k} - q_{\ell k}). \quad (115)
$$

Using (107)–(114), the Gaussian exponent for each replica $u$ and each token pair $(a, b)$ is therefore exactly

$$
-\frac{1}{2}\sum_{\ell=1}^{L}\sum_{k=1}^{L}\big(h_{ab}^{(\ell)\,u} - \omega_{ab}^{(\ell)}\big)[V^{-1}]_{\ell k}\big(h_{ab}^{(k)\,u} - \omega_{ab}^{(k)}\big) - \frac{1}{2}\sum_{r=1}^{L}(\eta_{ab}^{(r)})^2, \quad (116)
$$

which is the desired form with $\omega$ and $V$ identified as in (115).

Define the auxiliary measure

$$\mathcal{D}\eta = \prod_{a \leq b} \prod_{\ell=1}^{L} \frac{d\eta_{ab}^{(\ell)}}{\sqrt{2\pi}} \exp\left[-\frac{(\eta_{ab}^{(\ell)})^2}{2}\right]. \tag{117}$$

For fixed $\eta$, the replicas are independent and identically distributed through the conditional Gaussian $\mathcal{N}(h_{ab}; \omega_{ab}, V)$ on layers $\ell$. Hence, introducing the single-replica output partition function

$$\mathcal{Z}_{\text{out}}(y, \omega, V) = \int \prod_{a \leq b} d^L h_{ab} \, \mathcal{N}(h_{ab}; \omega_{ab}, V) \, P_{\text{out}}\left(y \mid \left\{\frac{h_{ab}^{(\ell)}}{\sqrt{2 - \delta_{ab}}}\right\}_{a \leq b, \, \ell=1}^{L}\right), \tag{118}$$

we can write

$$I_{\text{out}}(q) = \left[\int dy \int \mathcal{D}\eta \left(\mathcal{Z}_{\text{out}}(y, \omega, V)\right)^{m+1}\right]^n, \tag{119}$$

where $\omega$ and $V$ are the functions of $(q, Q)$ defined in (115). Expanding Eq. (119) for small number of replicas $m \to 0$ and using $A^{m+1} = A \, e^{m \ln A} = A \left(1 + m \ln A + o(m)\right)$, the contribution to the free entropy is governed by

$$I_{\text{out}}(q) = \int dy \int \mathcal{D}\eta \, \mathcal{Z}_{\text{out}}(y, \omega, V) \ln \mathcal{Z}_{\text{out}}(y, \omega, V), \tag{120}$$

We aim to compute $\partial_q I_{\text{out}}(q)$ to finally reach the state equation in Eq. (86), namely:

$$\widehat{q}_{\ell k} = 4\alpha \int dy \int \mathcal{D}\eta \left[1 + \ln \mathcal{Z}_{\text{out}}(y, \omega, V)\right] \frac{\partial \mathcal{Z}_{\text{out}}(y, \omega, V)}{\partial q_{\ell k}}. \tag{121}$$

It is convenient to rewrite the $\eta$-integral as an integral over $\omega$ itself. For each token pair $(a, b)$ we have $\omega_{ab} \sim \mathcal{N}(0, 2q)$ with independent draws across $a \leq b$. Thus we may equivalently write

$$I_{\text{out}}(q) = \int dy \int \left[\prod_{a \leq b} d^L \omega_{ab} \mathcal{N}(\omega_{ab}; 0, 2q)\right] \mathcal{Z}_{\text{out}}(y, \omega, V) \ln \mathcal{Z}_{\text{out}}(y, \omega, V), \tag{122}$$

where $\omega = \{\omega_{ab}^{(\ell)}\}_{a \leq b, \ell}$. Now define for each $(a, b)$ and layer $\ell$:

$$(g_{\text{out}}(y, \omega, V))_{ab}^{(\ell)} := \frac{\partial}{\partial \omega_{ab}^{(\ell)}} \ln \mathcal{Z}_{\text{out}}(y, \omega, V). \tag{123}$$

In (122), $q$ appears in: (i) in the Gaussian measure $\omega_{ab} \sim \mathcal{N}(0, 2q)$, and (ii) in $V = 2(Q - q)$ inside $\mathcal{Z}_{\text{out}}$. Let

$$F(\omega, q) := \int dy \, \mathcal{Z}_{\text{out}}(y, \omega, V) \ln \mathcal{Z}_{\text{out}}(y, \omega, V).$$

Then

$$I_{\text{out}}(q) = \int \left[\prod_{a \leq b} d^L \omega_{ab} \mathcal{N}(\omega_{ab}; 0, 2q)\right] F(\omega, q). \tag{124}$$

Now fix a single pair $(a, b)$, under $\omega_{ab} \sim \mathcal{N}(0, \Sigma)$ with $\Sigma = 2q$, we get the Gaussian identity, for any smooth $G(\omega_{ab})$:

$$\frac{\partial}{\partial \Sigma_{\ell k}} \mathbb{E}_{\omega_{ab} \sim \mathcal{N}(0, \Sigma)}[G(\omega_{ab})] = \frac{1}{2} \mathbb{E}_{\omega_{ab}}\left[\frac{\partial^2 G(\omega_{ab})}{\partial \omega_{ab}^{(\ell)} \partial \omega_{ab}^{(k)}}\right]. \tag{125}$$

Since $\Sigma = 2q$, we have $\partial/\partial q_{\ell k} = 2 \, \partial/\partial \Sigma_{\ell k}$, hence

$$\frac{\partial}{\partial q_{\ell k}} \mathbb{E}_{\omega_{ab} \sim \mathcal{N}(0, 2q)}[G(\omega_{ab})] = \mathbb{E}_{\omega_{ab}}\left[\frac{\partial^2 G(\omega_{ab})}{\partial \omega_{ab}^{(\ell)} \partial \omega_{ab}^{(k)}}\right]. \tag{126}$$

Applying (126) to (124) (and summing over independent pairs) yields

$$\frac{\partial I_{\text{out}}(q)}{\partial q_{\ell k}} = \int \left[\prod_{a \leq b} d^L \omega_{ab} \mathcal{N}(\omega_{ab}; 0, 2q)\right] \left(\sum_{a \leq b} \frac{\partial^2 F(\omega, q)}{\partial \omega_{ab}^{(\ell)} \partial \omega_{ab}^{(k)}} + \frac{\partial F(\omega, q)}{\partial q_{\ell k}}\bigg|_{\omega}\right), \tag{127}$$

where $\partial F/\partial q|_\omega$ is the explicit derivative through $V = 2(Q - q)$ at fixed $\omega$.

Since $V^{(rs)} = 2(Q_{rs} - q_{rs})$, we have $\partial V^{(rs)}/\partial q_{\ell k} = -2\delta_{r\ell}\delta_{sk}$, hence

$$\left.\frac{\partial F(\omega, q)}{\partial q_{\ell k}}\right|_\omega = -2\,\frac{\partial F(\omega, q)}{\partial V^{(\ell k)}}. \tag{128}$$

Plugging (128) into (127) gives

$$\frac{\partial I_{\text{out}}(q)}{\partial q_{\ell k}} = \int \left[\prod_{a \leq b} d^L \omega_{ab}\, \mathcal{N}(\omega_{ab}; 0, 2q)\right] \left(\sum_{a \leq b} \frac{\partial^2 F}{\partial \omega_{ab}^{(\ell)}\, \partial \omega_{ab}^{(k)}} - 2\,\frac{\partial F}{\partial V^{(\ell k)}}\right). \tag{129}$$

Now fix $\omega, V$ and define for brevity

$$Z(y) := \mathcal{Z}_{\text{out}}(y, \omega, V), \qquad g_{ab}^{(\ell)}(y) := (g_{\text{out}}(y, \omega, V))_{ab}^{(\ell)}. \tag{130}$$

Then

$$F(\omega, q) = \int dy\, Z(y)\, \ln Z(y). \tag{131}$$

We compute the two derivatives in (129) exactly. Since $Z$ is an integral of a Gaussian density $\mathcal{N}(h_{ab}; \omega_{ab}, V)$, then by definition:

$$\frac{\partial Z(y)}{\partial \omega_{ab}^{(\ell)}} = Z(y)\, g_{ab}^{(\ell)}(y), \qquad \frac{\partial^2 Z(y)}{\partial \omega_{ab}^{(\ell)}\, \partial \omega_{ab}^{(k)}} = Z(y)\, g_{ab}^{(\ell)}(y)\, g_{ab}^{(k)}(y) + Z(y)\, \frac{\partial g_{ab}^{(\ell)}(y)}{\partial \omega_{ab}^{(k)}}. \tag{132}$$

Therefore,

$$\frac{\partial F}{\partial \omega_{ab}^{(k)}} = \int dy\, (1 + \ln Z(y))\, \frac{\partial Z(y)}{\partial \omega_{ab}^{(k)}} = \int dy\, (1 + \ln Z(y))\, Z(y)\, g_{ab}^{(k)}(y), \tag{133}$$

$$\frac{\partial^2 F}{\partial \omega_{ab}^{(\ell)}\, \partial \omega_{ab}^{(k)}} = \int dy\, \left[Z(y)\, g_{ab}^{(\ell)}(y)\, g_{ab}^{(k)}(y) + (1 + \ln Z(y))\Big(Z(y)\, g_{ab}^{(\ell)}(y)\, g_{ab}^{(k)}(y) + Z(y)\, \partial_{\omega_{ab}^{(k)}} g_{ab}^{(\ell)}(y)\Big)\right]. \tag{134}$$

For a Gaussian integral, the derivative of $\ln Z$ with respect to $V$ satisfies the identity

$$\frac{\partial \ln Z(y)}{\partial V^{(\ell k)}} = \frac{1}{2}\left(g_{ab}^{(\ell)}(y)\, g_{ab}^{(k)}(y) + \frac{\partial g_{ab}^{(\ell)}(y)}{\partial \omega_{ab}^{(k)}}\right), \tag{135}$$

Using $\partial_{V^{(\ell k)}} Z = Z\, \partial_{V^{(\ell k)}} \ln Z$ and (135), we obtain

$$\frac{\partial F}{\partial V^{(\ell k)}} = \int dy\, (1 + \ln Z(y))\, \frac{\partial Z(y)}{\partial V^{(\ell k)}} = \frac{1}{2} \int dy\, (1 + \ln Z(y))\, Z(y)\, \left(g_{ab}^{(\ell)}(y)\, g_{ab}^{(k)}(y) + \partial_{\omega_{ab}^{(k)}} g_{ab}^{(\ell)}(y)\right). \tag{136}$$

Subtracting $2\, \partial F/\partial V^{(\ell k)}$ from $\partial^2 F/(\partial \omega_{ab}^{(\ell)}\, \partial \omega_{ab}^{(k)})$ using (134) and (136), all terms proportional to $(1 + \ln Z)Z(\cdots)$ cancel exactly, yielding

$$\frac{\partial^2 F}{\partial \omega_{ab}^{(\ell)}\, \partial \omega_{ab}^{(k)}} - 2\,\frac{\partial F}{\partial V^{(\ell k)}} = \int dy\, Z(y)\, g_{ab}^{(\ell)}(y)\, g_{ab}^{(k)}(y). \tag{137}$$

Plugging (137) into (129) and summing over $a \leq b$ gives

$$\frac{\partial I_{\text{out}}(q)}{\partial q_{\ell k}} = \int \left[\prod_{a \leq b} d^L \omega_{ab}\, \mathcal{N}(\omega_{ab}; 0, 2q)\right] \sum_{a \leq b} \int dy\, \mathcal{Z}_{\text{out}}(y, \omega, V)\, (g_{\text{out}}(y, \omega, V))_{ab}^{(\ell)}\, (g_{\text{out}}(y, \omega, V))_{ab}^{(k)}. \tag{138}$$

Therefore the output-channel state equation is

$$\widehat{q}_{\ell k} = 4\alpha\, \mathbb{E}_{\omega, y}\left[\sum_{a \leq b} (g_{\text{out}}(y, \omega, V))_{ab}^{(\ell)}\, (g_{\text{out}}(y, \omega, V))_{ab}^{(k)}\right], \tag{139}$$

which is equivalent to

$$\widehat{q}_{\ell k} = 4\alpha\, \mathbb{E}_{\eta, y}\left[\sum_{a \leq b} (g_{\text{out}}(y, \omega, V))_{ab}^{(\ell)}\, (g_{\text{out}}(y, \omega, V))_{ab}^{(k)}\right], \tag{140}$$

$$\text{for} \quad \ell = 1, \ldots, L \quad a \leq b = 1, \ldots, T.$$

The expectation $\mathbb{E}_{(\eta,y)}$ is taken over the joint measure

$$\prod_{\ell=1}^{L} \mathcal{D}\eta^{(\ell)} \tag{141}$$

and the output $y$ is drawn from the channel density

$$P_{\text{out}}\left(y \mid \{h_{ab}^{(\ell)}\}_\ell\right), \quad h_{ab}^{(\ell)} \sim \mathcal{N}\left(\omega_{ab}^{(\ell)}, V_{ab}^{(\ell k)}\right), \tag{142}$$

with:

$$P_{\text{out}}\left(y \mid \{h_{ab}^{(\ell)}\}_\ell\right) = \delta(\{y_{ab} - g(\{h_{ab}^{(\ell)}\}_{\forall \ell})_{ab}\}_{\forall ab}), \tag{143}$$

or particularly, for the deep attention case:

$$P_{\text{out}}\left(y \mid \{h_{ab}^{(\ell)}\}_\ell\right) = \delta(\{y_{ab} - B_c^L(\{h_{ab}^{(\ell)}\}_{\forall \ell})_{ab}\}_{\forall ab}). \tag{144}$$

### C.3.1 Recap of the state equations

In this section we summarize the findings of the previous appendices. We performed the Bayes optimal analysis of the attention-indexed models (AIM) defined in Eq. (1): we found that the problem can be split in two components, the former involving the (extensive width) rotationally invariant prior channel and the latter involving the output channel part of the model. Through a replica analysis, we found that the prior channel is described by the following function:

$$\mathcal{Z}_{\text{in}}(\{Y_\ell\}_{\ell=1}^L; \hat{q}) = \int \left[ \prod_{\ell=1}^L dS_\ell \, P_S(S_\ell) \right]$$
$$\times \exp\left[ -\frac{d}{4} \sum_{\ell,k=1}^L \hat{q}_{\ell k} \operatorname{Tr}(S_\ell S_k) + \frac{d}{2} \sum_{\ell,k=1}^L \sqrt{\hat{q}}_{\ell k} \operatorname{Tr}(Y_\ell S_k) \right]. \tag{145}$$

The denoising function associated to the prior channel assumes the form:

$$g_{\text{in}}(Y|\hat{q}) = \partial_{\hat{q}^{1/2}Y} \ln \mathcal{Z}_{\text{in}}(Y, \hat{q}) \tag{146}$$

The free entropy contribution coming from the prior channel is:

$$I_{\text{in}}(\hat{q}) = \lim_{d \to \infty} \frac{1}{d^2} \int DY_1 \dots DY_L \, \mathcal{Z}_{\text{in}}(Y_1, \dots, Y_L; \hat{q}) \log \mathcal{Z}_{\text{in}}(Y_1, \dots, Y_L; \hat{q}) \tag{147}$$

where $DY$ stands for integration over a $\text{GOE}(d)$ (Wigner) matrix $Y$.

Notably, this problem is equivalently mapped to the same posterior distribution of the following matrix denoising problem:

$$Y(S, \Delta)_\ell = S_\ell + \sum_{m=1}^L \sqrt{\Delta}_{\ell m} \Xi_m, \quad \Xi_m \sim \text{GOE}(d) \quad S_\ell \sim P_S \tag{148}$$

On the other hand, the output channel of the model is described by the function:

$$\mathcal{Z}_{\text{out}}(y, \omega, V) = \int \left[ \prod_{a \leq b}^T d^L h_{ab} \, \mathcal{N}(h_{ab}; \omega_{ab}; V_{ab}) \right] \delta\left( y - g(\{h_{ab}^{(\ell)}/\sqrt{2 - \delta_{ab}}\}_{\ell=1}^L) \right) \tag{149}$$
$$\text{with:} \quad \omega_{ab}^{(\ell)} = \sum_{k=1}^L \sqrt{2q}_{\ell k} \, \eta_{ab}^{(k)}, \qquad V^{(\ell k)} = 2(Q_{\ell k} - q_{\ell k})$$

The denoising function associated to the output channel takes the form:

$$g_{\text{out}}(y, \omega, V) = \partial_\omega \ln \mathcal{Z}_{\text{out}}(y, \omega, V) \tag{150}$$

The free entropy contribution coming from the output channel is:

$$I_{\text{out}}(q) = \int \prod_{a,b=1}^T dy_{ab} \int \mathcal{D}\eta_1 \dots \mathcal{D}\eta_L \, \mathcal{Z}_{\text{out}}(y, \omega, V) \log \mathcal{Z}_{\text{out}}(y, \omega, V) \tag{151}$$

where $\mathcal{D}\eta$ stands for integration over a $L \times T \times T$ tensor symmetric in the token indices and with independent entries $\mathcal{N}(0, 1)$.

To conclude this section, in the multi-layer setting described by the AIM framework in Eq. (1), we found the following state equations:

$$
\hat{q}_{\ell k} = 4\alpha \mathbb{E}_{\xi, \eta} \sum_{a \leq b}^{L} g_{\text{out}} \left( g\left( \left\{ \frac{h(\omega, V)_{ab}}{\sqrt{2 - \delta_{ab}}} \right\} \right), \omega, V \right)_{ab}^{(\ell)}
$$
$$
\times g_{\text{out}} \left( g\left( \left\{ \frac{h(\omega, V)_{ab}}{\sqrt{2 - \delta_{ab}}} \right\} \right), \omega, V \right)_{ab}^{(k)} , \tag{152}
$$
$$
q_{\ell k} = \lim_{d \to +\infty} \frac{1}{d} \mathbb{E}_{S, Y} \, \mathrm{Tr} \left[ g_{\text{in}}(Y(S, \hat{q}), \hat{q})_{\ell} g_{\text{in}}(Y(S, \hat{q}), \hat{q})_{k} \right] ,
$$

where $\mathbb{E}_{\eta, \xi}$ is intended as the average over $L \times T \times T$ symmetric in the token indices and Gaussians with zero mean and unit variance. Moreover, the average $\mathbb{E}_{S, Y}$ is with respect to respect to Y as given in Eq. (148) and $S \sim P_S$. Finally:

$$
[h(\omega, V)_{ab}]^{(\ell)} = \omega_{ab}^{(\ell)} + \sum_k \sqrt{V}^{(\ell k)} \xi_{ab}^{(k)} \tag{153}
$$

## C.4 The fixed point of AMP is described by the state equations

We start by defining a new variable $\omega_{\mu, ab}^*$ such that $y_\mu = g(\{\omega_{\mu, ab}^*/\sqrt{2 - \delta_{ab}}\}_{a \leq b})$, where we can assume that $\mathbb{E}[(\omega_{\mu, ab}^*)_\ell (\omega_{\mu, ab}^*)_k] = 2Q_{\ell k}^t$. Our first step is to define the quantities $m^t$ and $q^t$ on the iterates of AMP

$$
m_{\ell k}^t = \mathrm{Tr}[\hat{S}_\ell^t S_k^*]/d, \qquad q_{\ell k}^t = \mathrm{Tr}[\hat{S}_\ell^t \hat{S}_k^t]/d . \tag{154}
$$

We now claim, in analogy with [44, 35, 40, 42] that for every sample $\mu$ and every couple of tokens $a \leq b$ the variables $\omega_{\mu, ab}^t$ at each time converge to independent centered Gaussian variables with the following covariances

$$
\mathbb{E}[(\omega_{\mu, ab}^t)_\ell (\omega_{\mu, ab}^t)_k] = 2q_{\ell k}^t , \qquad \mathbb{E}[(\omega_{\mu, ab}^*)_\ell (\omega_{\mu, ab}^t)_k] = 2m_{\ell k}^t , \tag{155}
$$

By Nishimori's identities [50] we can assume $m^t = q^t$. The first equation of (21) is now immediately recovered (modulo the substitution $V \to 2(Q - q^t)$ which will come after)

$$
\hat{q}_{\ell k}^t \approx 4\alpha \mathbb{E}_{y, \omega^t} \sum_{a \leq b}^{T} \left[ g_{\text{out}}(y, \omega^t, V^t)_{ab}^{(\ell)} g_{\text{out}}(y, \omega^t, V^t)_{ab}^{(k)} \right] \tag{156}
$$

where

$$
y = g\left( \left\{ \frac{\omega_{ab}^*}{\sqrt{2 - \delta_{ab}}} \right\}_{a \leq b}^T \right) , \qquad \begin{pmatrix} \omega_{ab}^t \\ \omega_{ab}^* \end{pmatrix} \sim \mathcal{N}\left( 0, \begin{pmatrix} 2q^t & 2m^t \\ 2m^t & 2Q \end{pmatrix} \right) \tag{157}
$$

Again as in [44, 35, 40, 42] we will have that in distribution

$$
R_{ij}^t = S_{ij}^* + (\hat{q}^t)^{-1} \Xi_{ij}^t \tag{158}
$$

We are ready to close the circle: going back to the definition of $q^t$ we write

$$
q_{\ell k}^t = \mathbb{E}_{R^t} \, \mathrm{Tr} \left[ g_{\text{in}} \left( R^t, \hat{q}^t \right)_\ell g_{\text{in}} \left( R^t, \hat{q}^t \right)_k \right] /d , \tag{159}
$$

which is exactly the second equation in (21). Notice how the expectation is taken over the random variable $R_{ij}^t$ in (158). The last step is to notice that

$$
\hat{C}_{\ell k}^t = \mathrm{Tr}[(\hat{S}_\ell^t - S_\ell^*)(\hat{S}_k^t - S_k^*)]/d^2 = Q - q^t \tag{160}
$$

such that $V^t = 2(Q - q^t)$.

# D    The case of $L = 1$ **layer**

In this Appendix we restrict the theoretical results derived for an arbitrary number of layers to the particular case of $L = 1$ layer. In this particular case, the order parameters $q$ and $\hat{q}$ become scalar quantities. Moreover, in the following analysis we specialize to the extensive-rank choice:

$$S = \frac{1}{\sqrt{rd}} W W^\top \in \mathbb{R}^{d \times d} \quad W \in \mathbb{R}^{d \times r} \quad (W)_{ij} \sim \mathcal{N}(0, 1) \tag{161}$$

with rank ratio $\rho = r/d = O(1)$. Thus, the spectral distribution of the symmetric matrix S is that of the Marcenko-Pastur law for Wishart matrices described in App. (A).

## D.1    Prior channel state equation

Starting from Eq. (95) for $L = 1$ layer, we get::

$$I_{\text{in}}(\hat{q}) = \lim_{d \to \infty} \frac{1}{d^2} \int DY \, \mathcal{Z}_{\text{in}}(Y; \hat{q}) \log \mathcal{Z}_{\text{in}}(Y; \hat{q})$$

$$\mathcal{Z}_{\text{in}}(Y; \hat{q}) = \int dS \, P_0(S) \exp\left( -\frac{d}{2}\left(\hat{Q} + \frac{\hat{q}}{2}\right) \text{Tr}\left(S^T S\right) + \frac{\sqrt{\hat{q}}d}{2} \text{Tr}\left(Y^T S\right) \right). \tag{162}$$

Again, at the 0-replica order $\hat{Q} = 0$ and integrating over $Y$:

$$\int DY \, \mathcal{Z}_{\text{in}}(Y; \hat{q}) \tag{163}$$

$$= \int DY \int dS \, P_0(S) \exp\left( -\frac{\hat{q}d}{4} \text{Tr}(S^\top S) + \frac{\sqrt{\hat{q}}d}{2} \text{Tr}\left(Y^\top S\right) - \frac{1}{4} \text{Tr}\left(Y^\top Y\right) \right)$$

$$= \int dS \, P_0(S) \exp\left( \frac{\hat{q}d}{4} \text{Tr}\left(S^\top S\right) \right) \exp\left( -\frac{\hat{q}d}{4} \text{Tr}\left(S^\top S\right) \right)$$

$$= \int dS \, P_0(S) = 1 \tag{164}$$

Now, note that the exponent in $\mathcal{Z}_{\text{in}}(Y; \hat{q})$ can be rearranged as:

$$-\tfrac{\hat{q}d}{4} \text{Tr}(S^T S) + \frac{\sqrt{\hat{q}}d}{2} \text{Tr}(S^T Y) = -\frac{d}{4} \text{Tr}\left(\hat{q} S^T S - 2\sqrt{\hat{q}} S^T Y\right). \tag{165}$$

Observe

$$\text{Tr}\left[ (\sqrt{\hat{q}} S - Y)^T (\sqrt{\hat{q}} S - Y) \right] = \hat{q} \, \text{Tr}(S^T S) - 2\sqrt{\hat{q}} \, \text{Tr}(S^T Y) + \text{Tr}(Y^T Y). \tag{166}$$

Hence

$$-\tfrac{\hat{q}}{4} \text{Tr}(S^T S) + \frac{\sqrt{\hat{q}}}{2} \text{Tr}(Y^T S) = -\tfrac{1}{4} \text{Tr}\left[ (\sqrt{\hat{q}} S - Y)^2 \right] + \tfrac{1}{4} \text{Tr}(Y^T Y). \tag{167}$$

Therefore:

$$I_0(Y) = \exp\left[ +\tfrac{1}{4} \text{Tr}(Y^T Y) \right] \times \int dS \, P_0(S) \exp\left[ -\tfrac{1}{4} \text{Tr}\left(\sqrt{\hat{q}} S - Y\right)^2 \right]. \tag{168}$$

Ignoring the factor $\exp(\tfrac{1}{4} \text{Tr}(Y^T Y))$ that is independent of $S$, we see that

$$\int dS \, P_0(S) \exp\left[ -\tfrac{d}{4} \text{Tr}\left(\sqrt{\hat{q}} S - Y\right)^2 \right] \tag{169}$$

which plays the role of a posterior density for $S$ given $Y = \sqrt{\hat{q}} S + Z$ with $Z$ a GOE($d$) noise.

In the large-$d$ limit, let us parametrize $S$ by its eigenvalues:

$$S = U \Lambda U^T \tag{170}$$

where $\Lambda = \mathrm{diag}(\lambda_1, \ldots, \lambda_d)$. Then

$$dS = \left[\prod_{i=1}^{d} d\lambda_i\right] |\Delta(\{\lambda_i\})| \, dU \quad \text{with} \quad \Delta(\{\lambda_i\}) = \prod_{1 \leq i < j \leq d} |\lambda_i - \lambda_j|, \tag{171}$$

Then the exponent

$$\mathrm{Tr}\left[-\tfrac{1}{4}\left(\sqrt{\hat{q}}\,S - Y\right)^2\right] \tag{172}$$

becomes

$$-\tfrac{1}{4}\mathrm{Tr}\left(\sqrt{\hat{q}}\,U\,\Lambda\,U^T - Y\right)^2. \tag{173}$$

We can factor out the integral over $U \in \mathcal{O}(d)$ and for d large:

$$\int_{\mathcal{O}(d)} \exp\left(\tfrac{\hat{q}\,d}{2}\,\mathrm{Tr}[\Lambda\,U^T\,Y\,U]\right) \mathcal{D}U \;\approx\; \exp\left[\tfrac{d^2}{2}\,I_{\mathrm{HCIZ}}\big(\hat{q};\,\mu_\Lambda,\,\mu_Y\big)\right], \tag{174}$$

where $I_{\mathrm{HCIZ}}$ is an explicit functional in the limit $d \to \infty$ of dimension $2/d^2$ times the log of that integral, and $\mu_\Lambda$ is the limiting spectral distribution of $\Lambda/\sqrt{d}$.

The prior contribution of the free entropy is given by

$$\Phi_{\mathrm{prior}}(\hat{q}) = \lim_{d \to \infty} \frac{1}{d^2}\,\mathbb{E}\big[\ln I_0(Y)\big], \tag{175}$$

or more explicitly:

$$\Phi_{\mathrm{prior}}(\hat{q}) = \lim_{d \to \infty} \frac{1}{d^2}\,\mathbb{E}\left[\ln \int P_0(S)\,e^{-\frac{d}{4}\,\mathrm{Tr}(\sqrt{\hat{q}}\,S - Y)^2}\,dS\right]. \tag{176}$$

This term can be explicitly computed and mapped to a matrix estimation problem. i.e. a denoising problem as follows:

$$\Phi_{\mathrm{prior}}(\hat{q}) = \lim_{d \to \infty} \frac{1}{d^2}\,\mathbb{E}_Y \ln I_0(Y) = -\frac{\hat{q}\,Q}{4} + \frac{1}{2}I_{\mathrm{HCIZ}}\left(\hat{q};\mu_0,\mu_0 \boxplus \sigma_{\mathrm{sc},1/\sqrt{\hat{q}}}\right) + \text{const}, \tag{177}$$

where $Q = 1 + \rho$. Then, one has the relation from [51]:

$$-\frac{1}{2}\Sigma(\mu_{\hat{q}}) + \frac{1}{4\hat{q}}\mathbb{E}_{\mu_{\hat{q}}}[X^2] - \frac{1}{2}I_{\mathrm{HCIZ}}\left(\hat{q};\mu_0,\mu_{\hat{q}}\right) - \frac{3}{8} + \frac{1}{4}\ln \hat{q} + \frac{1}{4\hat{q}}\mathbb{E}_{\mu_0}[X^2] = 0, \tag{178}$$

where we have defined $\mu_{\hat{q}} = \mu_0 \boxplus \sigma_{\mathrm{sc},1/\sqrt{\hat{q}}}$ and $\Sigma(\mu)$ is the noncommutative entropy:

$$\Sigma(\mu) = \int \mu(dx)\mu(dy)\ln|x - y|.$$

In our normalization (with $Q = 1 + \rho$), rearranging yields

$$\frac{1}{2}I_{\mathrm{HCIZ}}(\hat{q};\mu_0,\mu_{\hat{q}}) = -\frac{1}{2}\Sigma(\mu_{\hat{q}}) + \frac{1}{4}\left[2Q\hat{q} + 1\right] - \frac{3}{8} - \frac{1}{4}\ln \hat{q}. \tag{179}$$

Plugging back into the free entropy, we obtain

$$\Phi_{\mathrm{prior}}(\hat{q}) = -\frac{\hat{q}\,Q}{4} + \left[-\frac{1}{2}\Sigma(\mu_{\hat{q}}) + \frac{1}{4}(2Q\hat{q} + 1) - \frac{3}{8} - \frac{1}{4}\ln \hat{q}\right] + \text{const}. \tag{180}$$

Taking the derivative with respect to $\hat{q}$ yields the "prior state" equation. In fact, differentiating we obtain

$$\frac{\partial \Phi}{\partial \hat{q}} = -\frac{q}{4} + \frac{Q}{4} - \frac{1}{4\hat{q}} - \frac{1}{2}\frac{\partial}{\partial \hat{q}}\Sigma(\mu_{\hat{q}}) = 0. \tag{181}$$

Using the derivative:

$$\frac{\partial}{\partial \hat{q}}\Sigma(\mu_{\hat{q}}) = -\frac{2\pi^2}{3\hat{q}^2}\int \mu_{\hat{q}}(x)^3\,dx, \tag{182}$$

this condition becomes

$$-\frac{q}{4} + \frac{Q}{4} - \frac{1}{4\hat{q}} + \frac{\pi^2}{3\hat{q}^2}\int \mu_Y(x)^3\,dx = 0. \tag{183}$$

which is exactly our desired state equation.

To sum up, in the problem

$$Y = \sqrt{\hat{q}}\, S + Z, \quad Z \sim \mathrm{GOE}(d), \tag{184}$$

the law of $Y$ is asymptotically $\mu_S \boxplus \sigma_{\mathrm{sc},\,1/\sqrt{\hat{q}}}$. we finally get:

$$q = Q - \frac{1}{\hat{q}} + \frac{4\,\pi^2}{3\,\hat{q}^2} \int \left[\mu_Y(x)\right]^3 \mathrm{d}x, \tag{185}$$

with $\mu_Y = \mu_S \boxplus \sigma_{\mathrm{sc},\,1/\sqrt{\hat{q}}}$.

For the computation of $\mu_Y$, we recall that if $\mu_Y = \mu_S \boxplus \sigma_{\mathrm{sc},\alpha}$, we can write

$$\mathcal{R}_{\mu_Y}(z) = \mathcal{R}_{\mu_S}(z) + \mathcal{R}_{\sigma_{\mathrm{sc},\,\alpha}}(z). \tag{186}$$

For the semicircle of radius $\alpha$, we have $\mathcal{R}_{\sigma_{\mathrm{sc},\,\alpha}}(z) = \alpha^2 z$. For $\mu_S$ (Marchenko–Pastur distribution with parameter $\rho$), we have

$$\mathcal{R}_{\mu_{MP,\rho}}(z) = \frac{\rho}{\sqrt{\rho}-z}. \tag{187}$$

In our case $\alpha = 1/\sqrt{\hat{q}}$, then

$$\mathcal{R}_{\mu_Y}(z) = \frac{\rho}{\sqrt{\rho}-z} + \alpha^2\, z. \tag{188}$$

From $\mathcal{R}_{\mu_Y}(z) = g_{\mu_Y}^{-1}(-z) - \frac{1}{z}$, one obtains an equation for $g_{\mu_Y}(z)$, with:

$$g_{\mu_Y}(z) = \int \frac{\mu_Y(\mathrm{d}x)}{x - z}. \tag{189}$$

So, using the identity $z = \frac{1}{x} + \mathcal{R}_{\mu_Y}(x)$, where $x = g_{\mu_Y}(z)$. So we get

$$z = \frac{1}{x} + \frac{\rho}{\sqrt{\rho}-x} + \alpha^2\, x. \tag{190}$$

Hence the final polynomial in $x$ is:

$$(\frac{1}{\sqrt{\rho}}\alpha^2)x^3 - (\frac{z}{\sqrt{\rho}} + \alpha^2)x^2 + (z + \frac{1}{\sqrt{\rho}} - \sqrt{\rho})x - 1 = 0 \iff x = g_{\mu_Y}(z). \tag{191}$$

We look for the solution of this equation with largest imaginary part. Moreover, we compute the discriminant of this third order equation in order to correctly quantify the edges of the spectral density we want to numerically compute.

Recalling $\alpha^2 = 1/\hat{q}$, the imaginary part of $x$ yields $\mu_Y$ (Stieltjes–Perron inversion), i.e.

$$\mu_Y(x_0) = \lim_{\epsilon \to 0^+} \frac{1}{\pi} \operatorname{Im} g_{\mu_Y}(x_0 - i\,\epsilon). \tag{192}$$

## D.2   Small/Large width limit of the prior channel

We recall that the state equations are of the form

$$Q - q = \frac{1}{\hat{q}} - \frac{4\pi^2}{3\hat{q}^2} \int \mathrm{d}x\, \mu_{1/\hat{q}}(x)^3$$
$$\hat{q} = 2\alpha F(Q - q, q) \tag{193}$$

where $\mu_{1/\hat{q}}$ is the spectral distribution of $S_* + \frac{1}{\sqrt{\hat{q}}}Z$ and $Q = d^{-1}\operatorname{Tr}(S_*^2)$. In our examples, $S^*$ is $\sqrt{\rho}$ times a standard Wishart, with $Q = 1 + \rho$.

### D.2.1 Small width limit

We follow [40, Section E.1.1]. Call $t = \rho/\hat{q}$, and $\bar{\alpha} = \alpha/\rho$. Call $\nu$ the distribution of $\sqrt{\rho}(S_* + \frac{1}{\sqrt{\hat{q}}}Z) = \sqrt{\rho}S_* + \sqrt{t}Z$, i.e.

$$\nu(y) = \rho^{-1/2}\mu_{1/\hat{q}}(\rho^{-1/2}y). \tag{194}$$

Notice that this is precisely the $\nu$ defined in [40, Eq. 56]. Then we have

$$
\begin{aligned}
Q - q &= \frac{1}{\hat{q}} - \frac{4\pi^2}{3\hat{q}^2}\int dx\,\mu_{1/\hat{q}}(x)^3 \\
&= \frac{t}{\rho} - \frac{4\pi^2 t^2}{3\rho^2}\rho\int dy\,[\rho^{-1/2}\mu_{1/\hat{q}}(\rho^{-1/2}y)]^3 \\
&= \frac{t}{\rho} - \frac{4\pi^2 t^2}{3\rho}\int dy\,\nu(y)^3 \\
&= \frac{t}{\rho}\left[1 - \frac{4\pi^2 t}{3}\int dy\,\nu(y)^3\right] \\
&\approx \begin{cases} t(2-t) & \text{if}\quad t \le 1 \\ 1 & \text{if}\quad t > 1 \end{cases}
\end{aligned} \tag{195}
$$

where we used [40, Eq. 57 and following] to take the limit of small $\kappa$ at leading order. Thus, the equations can be recast to

$$
\begin{aligned}
Q - q &= \begin{cases} t(2-t) & \text{if}\quad t \le 1 \\ 1 & \text{if}\quad t > 1 \end{cases} \\
t &= \frac{1}{2\bar{\alpha}F(Q-q)}.
\end{aligned} \tag{196}
$$

In particular, we have a weak recovery threshold. Indeed, as long as

$$\bar{\alpha} < \frac{1}{2F(1)} \tag{197}$$

we have that $Q - q = 1$, i.e. the same error as the average from the prior (BO estimator with no data).

### D.2.2 Large width limit

Recall that $Q = 1 + \rho$ and $q \in [\rho, 1 + \rho]$, so that $Q - q \in [0, 1]$ even in the $\rho \to \infty$ limit. Then we have

$$
\begin{aligned}
Q - q &= \frac{1}{\hat{q}} - \frac{4\pi^2}{3\hat{q}^2}\int dx\,\mu_{1/\hat{q}}(x)^3 \\
&= \frac{1}{\hat{q}}\left[1 - \frac{4\pi^2}{3\hat{q}\rho}\int dy\,[\sqrt{\rho}\mu_{1/\hat{q}}(\sqrt{\rho}y)]^3\right] \\
&= \frac{1}{\hat{q}}\left[1 - \frac{4\pi^2}{3\hat{q}\rho}\int dy\,\mu_{1/\rho\hat{q}}(y)^3\right] \\
&\approx \frac{1}{\hat{q}}\left[1 - \frac{1}{1+\hat{q}}\right] \\
&\approx \frac{1}{1+\hat{q}},
\end{aligned} \tag{198}
$$

where we used [40, Section E.2].

### D.3 Output channel state equation

For $L = 1$ layers we obtain the state equation for the output channel contribution:

$$\hat{q} = 4\,\alpha\,\mathbb{E}_{(\eta,y)}\left[\sum_{a\le b}(g_{\text{out}}(y,\omega,V))^2_{ab}\right] \tag{199}$$

Where we remind $\eta_{ab} \sim \mathcal{N}(0,1)$ with $a \leq b = 1, \ldots, T$ and $\omega_{ab} = \sqrt{2q}\,\eta_{ab}$, $V = 2(Q - q)$, $Q = 1 + \rho$. The denoising function is given by:

$$(g_{\text{out}}(y, \omega, V))_{ab} = \partial_{\omega_{ab}} \ln \mathcal{Z}_{\text{out}}(y, \omega, V) \tag{200}$$

and:

$$\mathcal{Z}_{\text{out}}(y, \omega, V) = \int \prod_{a \leq b} dh_{ab} \mathcal{N}(h_{ab}, \omega_{ab}, V)\,\delta\big(y - f(h)\big) \tag{201}$$

where $f(h)$ depends on the precise choice of the model. In particular, in the following sections we consider the three cases dealt in the main text. In the following, we first consider a linear output channel for a generic number of tokens $T$. This simple case serves as a baseline for the more interesting case of the softmax channel, namely the self-attention layer for an arbitrary number of tokens. We also consider the hardmax variant of the model treated in the main text for $T = 2$ tokens.

### D.4  Linear output channel for generic number of tokens

We consider $P_{\text{out}}(y_{ab} \mid h_{ab}) = \delta(y_{ab} - \frac{h_{ab}}{\sqrt{2 - \delta_{ab}}})$. Then

$$\mathcal{Z}_{\text{out}}(y, \omega, V) = \int \Big[ \prod_{a \leq b} \mathcal{N}(h_{ab}; \omega_{ab}, V) \Big] \prod_{a \leq b} \delta\Big[ y_{ab} - \frac{h_{ab}}{\sqrt{2 - \delta_{ab}}} \Big]\,dh. \tag{202}$$

Enforcing $h_{ab} = \sqrt{2 - \delta_{ab}}\,y_{ab}$ , this gives directly:

$$\mathcal{Z}_{\text{out}}(y, \omega, V) = \prod_{a \leq b} \Big[ \frac{1}{\sqrt{2\pi V}} \exp\Big( -\frac{(\sqrt{2 - \delta_{ab}}\,y_{ab} - \omega_{ab})^2}{2 V} \Big) \Big]. \tag{203}$$

Hence

$$\ln \mathcal{Z}_{\text{out}}(y, \omega, V) = \sum_{a \leq b} \Big[ -\tfrac{1}{2} \ln(2\pi V) \; - \; \frac{(\sqrt{2 - \delta_{ab}}\,y_{ab} - \omega_{ab})^2}{2 V} \Big]. \tag{204}$$

We recall that $\omega_{ab}(\eta)$ depends linearly on $\eta_{ab}$, e.g.:

$$\omega_{ab} = \sqrt{2q}\,\eta_{ab}, \quad V_{ab} = 2(Q - q), \quad Q = 1 + \rho. \tag{205}$$

Then

$$(g_{\text{out}}(y, \omega, V))_{ab} = \frac{\partial}{\partial \omega_{ab}} \ln \mathcal{Z}_{\text{out}}(y, \omega, V) = -\frac{\partial}{\partial \omega_{ab}} \Big[ \frac{(\sqrt{2 - \delta_{ab}}\,y_{ab} - \omega_{ab}(\eta))^2}{2 V} \Big] = + \frac{(\sqrt{2 - \delta_{ab}}\,y_{ab} - \omega_{ab})}{V}. \tag{206}$$

Thus

$$\sum_{a \leq b} (g_{\text{out}}(y, \omega, V))_{ab}^2 = \sum_{a \leq b} \Big( \big[ \sqrt{2 - \delta_{ab}}\,y_{ab} - \omega_{ab}(\eta) \big] \frac{1}{2(Q - q)} \Big)^2. \tag{207}$$

We can compute the expectation:

$$\mathbb{E}\Big[ \sum_{a \leq b} (g_{\text{out}}(y, \omega, V))_{ab}^2 \Big]$$

$$= \int \Big( \prod_{a \leq b} d\eta_{ab}\, \frac{e^{-\eta_{ab}^2/2}}{\sqrt{2\pi}} \Big) \int \Big( \prod_{a \leq b} dy_{ab} \Big) \mathcal{Z}_{\text{out}}(y, \omega, V) \sum_{a \leq b} (g_{\text{out}}(y, \omega, V))_{ab}^2. \tag{208}$$

We can simply use:

$$\int dy_{ab}\, \frac{1}{\sqrt{2\pi V}} \exp\Big( -\frac{(\sqrt{2 - \delta_{ab}}\,y_{ab} - \omega_{ab})^2}{2 V} \Big) \big[ \sqrt{2 - \delta_{ab}}\,y_{ab} - \omega_{ab} \big]^2 = V. \tag{209}$$

Therefore,

$$\int \Big( \prod_{a \leq b} dy_{ab} \Big) \mathcal{Z}_{\text{out}}(y, \omega, V) \sum_{a \leq b} (g_{\text{out}}(y, \omega, V))_{ab}^2 = \sum_{a \leq b} \Big[ \big( \frac{1}{2(Q - q)} \big)^2 V \Big]. \tag{210}$$

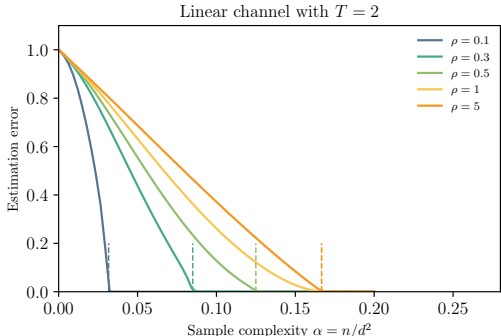

Figure 3: Illustration of the Bayes-optimal error for the linear output channel baseline in Eq. (202), for $T = 2$ tokens and several values of the width ratio $\rho = r/d$. The model reaches zero BO error at finite $\alpha$. The recovery threshold matches perfectly the one find by the simple counting argument in (215), plotted in short vertical lines.

Thus each term becomes

$$\left(\frac{1}{2(Q-q)}\right)^2 2(Q-q) = \frac{1}{2(Q-q)}. \tag{211}$$

Hence the entire sum is

$$\sum_{a \leq b} \frac{1}{2(Q-q)} = \frac{T(T+1)}{4} \frac{1}{Q-q}. \tag{212}$$

Notice that this result does not depend on $\eta$. Consequently, the outer integral over $\eta$ becomes 1.Hence we arrive to the final form of the linear output channel state equation:

$$\hat{q} = 4\alpha \, \mathbb{E}_{(\eta,y)}\left[\sum_{a \leq b}(g_{\text{out}}(y,\omega,V))_{ab}^2\right] = 4\alpha \sum_{a \leq b}\left[\mathbb{E}_{(\eta,y)}g_{\text{out}}(y,\omega,V))_{ab}^2\right] = 4\alpha\frac{T(T+1)}{4(Q-q)} \tag{213}$$

which finally simplifies into the output channel state equation:

$$\hat{q} = \frac{T(T+1)\,\alpha}{Q-q} \tag{214}$$

As an example, in Fig. 3 we show the fixed point solution for the state evolution equations for the linear output channel. The prior equation (26) remains unchanged, while we use Eq. (213) for simulating the linear output channel results. We also show in vertical dashed lines the recovery threshold found by the simple counting problem:

$$\frac{T(T+1)}{2}\alpha_{\text{count}} = \rho - \frac{\rho^2}{2} \tag{215}$$

The linear output channel matches perfectly the counting recovery threshold, unlike in the softmax case shown in Eq. (32).

### D.5 Softmax output channel for generic number of tokens

We compute the quantity:

$$\mathcal{Z}_{\text{out}}(y,\omega,V) = \int \prod_{a \leq b} dh_{ab} \frac{1}{\sqrt{2\pi V_{ab}}} e^{-\frac{(h_{ab}-\omega_{ab})^2}{2V_{ab}}} \prod_{a \leq b} \delta(y_{ab} - \text{Softmax}\{\frac{\beta}{\sqrt{2-\delta_{ab}}}h_{ab}\}). \tag{216}$$

where we remind the factor $\sqrt{2-\delta_{ab}}$ is present due to the symmetrization of the problem (i.e. multiply and divide by $\sqrt{2-\delta_{ab}}$), allowing a much simpler treatment of the BO analysis in change of this slight modification of the output channel.

From now on, we define the quantity $\tau_{ab} = \sqrt{2 - \delta_{ab}}$. We thus aim to compute the quantity:

$$\mathcal{Z}_{\text{out}}(y, \omega, V) = \int \prod_{a \leq b} dh_{ab} \delta(y - \sigma(\frac{h_{ab}}{\tau_{ab}})) \prod_{a \leq b} \mathcal{N}(h_{ab}, \omega_{ab}, V_{ab}) \tag{217}$$

We introduce the variable $z_{ab} = h_{ab}/\tau_{ab}$ and exploit $dh\mathcal{N}(h, \mu, \sigma) = dz\mathcal{N}(z, \mu/\tau, \sigma/\tau^2)$, we get:

$$\mathcal{Z}_{\text{out}}(y, \omega, V) = \int \prod_{a \leq b} dz_{ab} \, \delta(y - \sigma(z)) \prod_{a \leq b} \mathcal{N}(z_{ab}, \frac{\omega_{ab}}{\tau_{ab}}, \frac{V_{ab}}{\tau_{ab}^2}) =$$

$$= \int \prod_{a \leq b < T} dt_{ab} \, \mathcal{N}(t_{ab}, \frac{\omega_{ab}}{\tau_{ab}} - s_a, \frac{V_{ab}}{\tau_{ab}^2}) \prod_{a=1}^{T} ds_a \, \mathcal{N}(s_a, \frac{\omega_{aT}}{\tau_{aT}}, \frac{V_{aT}}{\tau_{aT}^2}) \tag{218}$$

where in the last equality we introduced the inverse mapping of the row-wise softmax function, defined in Eq. (40). In particular, we introduce:

$$e^{\beta t_{ab}} = \frac{e^{\beta z_{ab}}}{e^{\beta z_{aT}}} = \frac{e^{\beta z_{ab}}}{\sum_{b=1}^{T} e^{\beta z_{ab}}} \left( \frac{e^{\beta z_{aT}}}{\sum_{b=1}^{T} e^{\beta z_{ab}}} \right)^{-1} = \frac{y_{ab}}{y_{aT}} \quad \forall a \leq b < T \tag{219}$$

which leads to:

$$t_{ab} = \frac{1}{\beta} \log(\frac{y_{ab}}{y_{aT}}) = \phi_{ab}(y) \quad \forall a \leq b < T \tag{220}$$

while for $b = T$:

$$\frac{y_{Ta}}{y_{TT}} = \frac{e^{\beta z_{Ta}}}{e^{\beta z_{TT}}} = \frac{e^{\beta z_{aT}}}{e^{\beta z_{TT}}} = e^{\beta(s_a - s_{TT})} \rightarrow s_a = s_{TT} + \phi_{Ta}(y) \quad \forall a < T \tag{221}$$

having introduced the change of variables:

$$z_{ab} \rightarrow t_{ab} = z_{ab} - z_{aT} \rightarrow z_{ab} = t_{ab} + s_a = \phi_{ab} + \phi_{Ta} + s_{TT} \quad \forall a \leq b < T \tag{222}$$

and

$$z_{aT} \rightarrow s_a = z_{aT} \rightarrow z_{aT} = s_a = s_{TT} + \phi_{Ta} \quad \forall a < T \tag{223}$$

Having this mapping clear and introducing the short-hand notation $\tilde{\omega} = \omega/\tau$ and $\tilde{V} = V/\tau^2$, $s_{TT} = x$, we can see that we can reduce the computation of Eq. (218) to that of one simple scalar integral in the variable $x = s_T$, namely:

$$\mathcal{Z}_{\text{out}}(y, \omega, V) = \int dx \mathcal{N}(x, \tilde{\omega}_{TT}, \tilde{V}_{TT}) \prod_{a=1}^{T-1} \mathcal{N}(x + \phi_{Ta}(y), \tilde{\omega}_{aT}, \tilde{V}_{aT})$$

$$\times \prod_{a \leq b < T} \mathcal{N}(\phi_{ab}(y) + \phi_{Ta}(y) + x, \tilde{\omega}_{ab}, \tilde{V}_{ab}) \tag{224}$$

$$= \int dx \exp\left\{ -\frac{1}{2} \left[ \sum_{a=1}^{T} \frac{(x + \phi_{Ta} - \tilde{\omega}_{aT})^2)}{\tilde{V}_{aT}} + \sum_{a \leq b < T} \frac{(\phi_{ab} + \phi_{Ta} + x - \tilde{\omega}_{ab})^2}{\tilde{V}_{ab}} \right] \right\}$$

We thus obtain a simple gaussian integral whose exponential is of the form:

$$-\frac{1}{2} \left[ x^2 \left( \sum_{a \leq b} \tilde{V}_{ab}^{-1} \right) + 2x \left( \sum_{a=1}^{T} \frac{\phi_{Ta} - \tilde{\omega}_{aT}}{\tilde{V}_{aT}} + \sum_{a \leq b < T} \frac{\phi_{ab} + \phi_{Ta} - \tilde{\omega}_{aT}}{\tilde{V}_{ab}} \right) \right.$$

$$\left. + \left( \sum_{a=1}^{T} \frac{(\phi_{Ta} - \tilde{\omega}_{aT})^2}{\tilde{V}_{aT}} + \sum_{a \leq b < T} \frac{(\phi_{ab} + \phi_{Ta} - \tilde{\omega}_{ab})^2}{\tilde{V}_{ab}} \right) \right] \tag{225}$$

Having computed this simple gaussian integral, we can hence compute the quantity of interest:

$$\log \mathcal{Z} = \frac{1}{2\tilde{V}} \left[ \sum_{a=1}^{T} \frac{\phi_{Ta} - \tilde{\omega}_{aT}}{\tilde{V}_{aT}} + \sum_{a \leq b < T} \frac{\phi_{ab} + \phi_{Ta} - \tilde{\omega}_{aT}}{\tilde{V}_{ab}} \right]^2$$

$$- \frac{1}{2} \left[ \sum_{a=1}^{T} \frac{(\phi_{Ta} - \tilde{\omega}_{aT})^2}{\tilde{V}_{aT}} + \sum_{a \leq b < T} \frac{(\phi_{ab} - \phi_{Ta} - \tilde{\omega}_{ab})^2}{\tilde{V}_{ab}} \right] + \text{cost} \tag{226}$$

with $\tilde{V} = \sum_{a \leq b} \tilde{V}_{ab}^{-1}$ and again $\phi_{ab}(y) = \frac{1}{\beta} \log \frac{y_{ab}}{y_{aT}}$, $\tilde{\omega}_{ab} = \frac{\omega_{ab}}{\sqrt{2-\delta_{ab}}}$, $\tilde{V}_{ab} = \frac{Vab}{2-\delta_{ab}}$, $\omega_a = \sqrt{2q}\eta_{ab}$, $V_{ab} = V = 2(Q - q)$, $h \sim \mathcal{N}(\tilde{\omega}, \tilde{V})$, $y = \sigma(h)$. The constant term contains those terms independent from $\omega$, as we are finally interested in the denoising function , which is the derivative:

$$g_{\text{out}}(y, \omega, V)_{ab} = \partial_{\omega_{ab}} \log \mathcal{Z}_{\text{out}}(y, \omega, V) \tag{227}$$

We thus compute the denoising function deriving with quantity $\log \mathcal{Z}_{\text{out}}(y, \omega, V)$ with respect to $\tilde{\omega}$, thus computing $\tau_{ij} \partial_{\omega_{ij}} \log \mathcal{Z}_{\text{out}}(y, \omega, V)$ for $i \leq j < T$ and for $j = T$. We also consider that $\sum_{a \leq b}^{T} \tilde{V}_{ab}^{-1} = \frac{1}{V} \sum_{a \leq b}^{T}(2 - \delta_{ab}) = \frac{T^2}{V}$, $V = 2(Q - q)$.

Finally, we obtain the final form of the denoising function of the softmax output channel in Eq. (27) for an arbitrary number of tokens, substituting back the original $V$ and $\omega$:

$$V(g_{\text{out}})_{ij} = -\frac{\tau_{ij}}{T^2} \left[ \sum_{a \leq b}^{T} \tau_{ab}^2 \phi_{Ta} - \sum_{a \leq b}^{T} \tau_{ab}\omega_{ab} + \sum_{a \leq b}^{T-1} \tau_{ab}^2 \phi_{ab} \right] + \tau_{ij}\phi_{Ti} - \omega_{ij} + \delta(j < T)\phi_{ij}\tau_{ij} \tag{228}$$

which is exactly the same form appeared in the main text in Eq. (31).

We now complete this appendix by computing the quantity $\mathbb{E}_{\eta,y} \sum_{a \leq b}(g_{\text{out}})_{ab}^2$ . To do so, we exploit the following relations:

$$\phi_{ab} = h_{ab} - h_{aT} \quad h \sim \mathcal{N}(\tilde{\omega}, \tilde{V}) \rightarrow \tau_{ab}h_{ab} = \sqrt{2q}\,\eta_{ab} + \sqrt{V}\,\xi_{ab} \quad a \leq b \leq T \tag{229}$$

with $\eta_{ab}, \xi_{ab} \sim \mathcal{N}(0, 1)$ and

$$\phi_{Ta} = h_{Ta} - h_{TT} = h_{aT} - h_{TT} \tag{230}$$

We thus substitute these relationships inside Eq. (228) and finally compute $\mathbb{E}_{\eta,\xi} \sum_{a \leq b}(g_{\text{out}})_{ab}^2$. After a long but simple algebraic calculation, it is possible to show that the denoiser function reduces to simply:

$$\begin{aligned}
V(g_{\text{out}})_{ij} = &= \tau_{ij}\sqrt{V}\xi_{TT} - \frac{\tau_{ij}}{T^2} \sum_{a \leq b}^{T^2} \tau_{ab}\sqrt{V}\xi_{ab} + \frac{\tau_{ij}}{\tau_{iT}}\sqrt{V}\xi_{iT} \\
&\quad - \frac{\tau_{ij}}{\tau_{TT}}\sqrt{V}\xi_{TT} + \delta(j < T)\sqrt{V}\xi_{ij} - \delta(j < T)\frac{\tau_{ij}}{\tau_{iT}}\sqrt{V}\xi_{iT} \\
&= -\frac{\tau_{ij}}{T^2} \sum_{a \leq b}^{T^2} \tau_{ab}\sqrt{V}\xi_{ab} + \sqrt{V}\xi_{iT}\delta(j = T) + \delta(j < T)\sqrt{V}\xi_{ij} \\
&= -\frac{\tau_{ij}}{T^2} \sum_{a \leq b}^{T^2} \tau_{ab}\sqrt{V}\xi_{ab} + \sqrt{V}\xi_{ij}
\end{aligned} \tag{231}$$

which finally gives:

$$\begin{aligned}
\mathbb{E}_{\eta,\xi} V \sum_{i \leq j}^{T}(g_{\text{out}})_{ij}^2 &= \frac{T(T+1)}{2} - \frac{2}{T^2} \sum_{i \leq j}^{T} \sum_{a \leq b}^{T} \tau_{ij}\tau_{ab}\mathbb{E}\xi_{ij}\xi_{ab} \\
&\quad + \frac{1}{T^4} \sum_{i \leq j}^{T} \sum_{a \leq b}^{T} \sum_{c \leq d}^{T} \tau_{ij}^2 \tau_{ab}\tau_{cd}\mathbb{E}\xi_{ab}\xi_{cd} \\
&= \frac{T(T+1)}{2} - \frac{2}{T^2} \sum_{i \leq j}^{T} \tau_{ij}^2 + \frac{1}{T^4} \sum_{i \leq j}^{T} \sum_{a \leq b}^{T} \tau_{ij}^2 \tau_{ab}^2 \\
&= \frac{T^2 + T - 2}{2}
\end{aligned} \tag{232}$$

Hence, we can finally conclude that the output channel state equation we obtain for a self-attention layer with an arbitrary number of tokens is:

$$\hat{q} = 4\alpha \mathbb{E}_{\eta,\xi} \sum_{i \leq j}^{T}(g_{\text{out}})_{ij}^2 = \frac{4\alpha(T^2 + T - 2)}{2V} = \frac{\alpha(T^2 + T - 2)}{Q - q} \tag{233}$$

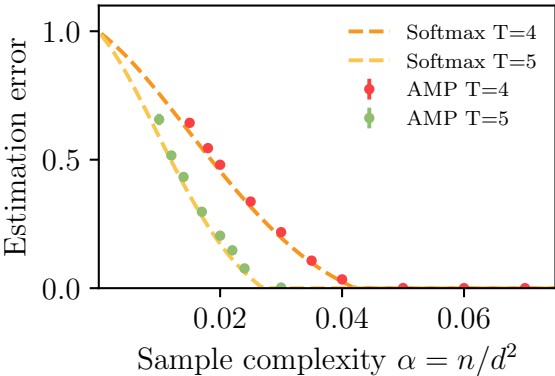

Figure 4: Comparison between the fixed points solutions of the state equations for a softmax output channel in Eq. (26) and Eq. (233) for $T = 4, 5$ tokens. We compare the theoretical solution with their corresponding AMP algorithm run over 16 different realizations and with $d = 120$. The error bars in the AMP dots are computed with respect to the mean value.

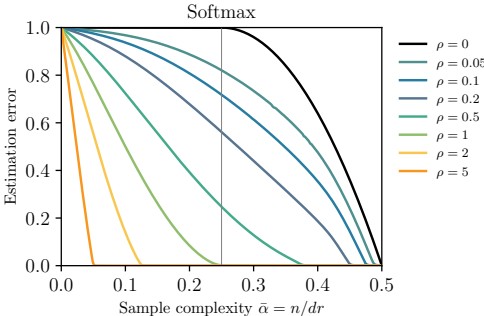

Figure 5: Low width limit of the self-attention model for $L = 1$ layer and $T = 2$ tokens in Eq. (27). We rescale the sample ratio as $\bar{\alpha} = n/dr$ and we plot several values of the width ratio $\rho = r/d$. We correctly predict the weak recovery threshold in Eq. (197).

which is the result presented in the main text in Eq. (32). We highlight this final result holds for any value of the softmax inverse temperature $0 < \beta < +\infty$. In Fig. 4 we show for completeness the state equations for the softmax output channel described by (216) for $T = 4, 5$ tokens and its corresponding AMP run for 16 different realizations and with $d = 120$. The error bars in the AMP dots are computed with respect to the mean value. We find a good agreement also in this case. In Fig. 5 we plot the low width behavior of the self-attention model for $L = 1$ layer and $T = 2$ tokens in Eq. (27), for which we recover the simple output channel state equation in Eq. (233), thus giving:

$$F(Q - q, q) = \frac{T^2 + T - 2}{2(Q - q)} \tag{234}$$

Regarding the large width result in Eq. (198) in this softmax output channel case, we get the equation:

$$\hat{q} = \frac{\alpha(T^2 - T + 2)}{Q - q} = \alpha(T^2 - T + 2)(1 + \hat{q}) \tag{235}$$

so we get

$$\hat{q} = \frac{\alpha(T^2 - T + 2)}{1 - \alpha(T^2 - T + 2)} \tag{236}$$

which gives the large $\rho$ result:

$$\text{MMSE} = \frac{1}{1 + \hat{q}} = 1 - \alpha(T^2 - T + 2). \tag{237}$$

### D.6 Hardmax output channel for 2 tokens

We now discuss the hardmax output channel case, in the special case of $T = 2$ tokens. Following Eq. (28) in the main text, we need to compute the quantity:

$$\mathcal{Z}_{\text{out}}(y, \omega, V) = \int \prod_{a \leq b} dh_{ab} \frac{1}{\sqrt{2\pi V_{ab}}} e^{-\frac{(h_{ab} - \omega_{ab})^2}{2V_{ab}}} \prod_{a \leq b} \delta(y_{ab} - \sigma_{\text{hard}}(\{\frac{1}{\sqrt{2 - \delta_{ab}}} h_{ab}\}_a)_b). \quad (238)$$

with:

$$\sigma_{\text{hard}}(z_1 \ldots z_T)_i = \delta(i = \arg\max_j x_j) \quad (239)$$

In this setting, in particular when $T = 2$, the output label of e.g. $y_{11}$ becomes

$$y_{11} = \Theta(h_{11} - h_{12}) \quad (240)$$

and similarly for the other labels. Here $\Theta(u)$ is the Heaviside function:

$$\Theta(u) = \begin{cases} 1, & u > 0, \\ 0, & u < 0. \end{cases} \quad (241)$$

For the computation of the quantity in Eq. (238), it is convenient to make a change of variables by introducing the differences:

$$u = h_{11} - \frac{h_{12}}{\sqrt{2}}, \quad v = h_{22} - \frac{h_{12}}{\sqrt{2}}. \quad (242)$$

We can thus rewrite in the case of $T = 2$ tokens:

$$I_{\text{out}}(\eta, y) = \int dh_{12} \, \mathcal{N}(h_{12}; \omega_{12}, V_{12}) \left\{ \int_{u \in \mathcal{R}(y_{11})} du \, \mathcal{N}(u + \frac{h_{12}}{\sqrt{2}}; \omega_{11}, V_{11}) \right\}$$
$$\times \left\{ \int_{v \in \mathcal{R}(y_{22})} dv \, \mathcal{N}(v + \frac{h_{12}}{\sqrt{2}}; \omega_{22}, V_{22}) \right\}, \quad (243)$$

where the integration ranges are defined by the hard–threshold:

$$\mathcal{R}(y_{11}) = \begin{cases} \{u > 0\}, & \text{if } y_{11} = 1, \\ \{u < 0\}, & \text{if } y_{11} = 0, \end{cases} \quad \mathcal{R}(y_{22}) = \begin{cases} \{v > 0\}, & \text{if } y_{22} = 1, \\ \{v < 0\}, & \text{if } y_{22} = 0. \end{cases}$$

Now, by shifting the Gaussian factors we have

$$\mathcal{N}(u + h_{12}; \omega_{11}, V_{11}) = \mathcal{N}(u; \omega_{11} - \frac{h_{12}}{\sqrt{2}}, V_{11}), \quad (244)$$

and similarly for the $v$–integral. Thus, the expression becomes

$$\mathcal{Z}_{\text{out}}(y, \omega, V) = \int dh_{12} \, \mathcal{N}(h_{12}; \omega_{12}, V_{12}) \, F_{11}(\frac{h_{12}}{\sqrt{2}}; \omega) \, F_{22}(\frac{h_{12}}{\sqrt{2}}; \omega), \quad (245)$$

with

$$F_{11}(\frac{h_{12}}{\sqrt{2}}; \omega) = \int_{u \in \mathcal{R}(y_{11})} du \, \mathcal{N}(u; \omega_{11} - \frac{h_{12}}{\sqrt{2}}, V_{11}) = \Phi\left(s_{11} \frac{\omega_{11} - \frac{h_{12}}{\sqrt{2}}}{\sqrt{V_{11}}}\right), \quad (246)$$

$$F_{22}(\frac{h_{12}}{\sqrt{2}}; \omega) = \int_{v \in \mathcal{R}(y_{22})} dv \, \mathcal{N}(v; \omega_{22} - \frac{h_{12}}{\sqrt{2}}, V_{22}) = \Phi\left(s_{22} \frac{\omega_{22} - \frac{h_{12}}{\sqrt{2}}}{\sqrt{V_{22}}}\right), \quad (247)$$

where $\Phi(z)$ is the standard Gaussian CDF and

$$s_{11} = 2y_{11} - 1 = \begin{cases} +1, & y_{11} = 1, \\ -1, & y_{11} = 0, \end{cases} \quad s_{22} = 2y_{22} - 1 = \begin{cases} +1, & y_{22} = 1, \\ -1, & y_{22} = 0. \end{cases}$$

Thus, in the hard–threshold limit the output channel integral is given by:

$$\mathcal{Z}_{\text{out}}(y, \omega, V) = \int_{-\infty}^{+\infty} dh_{12} \, \mathcal{N}(h_{12}; \omega_{12}, V_{12}) \, \Phi\left(s_{11} \frac{\omega_{11} - \frac{h_{12}}{\sqrt{2}}}{\sqrt{V_{11}}}\right) \Phi\left(s_{22} \frac{\omega_{22} - \frac{h_{12}}{\sqrt{2}}}{\sqrt{V_{22}}}\right) \quad (248)$$

We can further manipulate this expression.

Writing $h_{12} = \omega_{12} + \sqrt{V_{12}}\, Z$ with $Z \sim \mathcal{N}(0,1)$; then, using independence,

$$\mathcal{Z}_{\text{out}}(y, \omega, V) = \mathbb{E}_Z\Big[\Phi\big(u_1 - \lambda_1 Z\big)\,\Phi\big(u_2 - \lambda_2 Z\big)\Big], \tag{249}$$

where

$$u_1 = s_{11}\,\frac{\sqrt{2}\omega_{11} - \omega_{12}}{\sqrt{2V_{11}}}, \quad u_2 = s_{22}\,\frac{\sqrt{2}\omega_{22} - \omega_{12}}{\sqrt{2V_{22}}}, \tag{250}$$

$$\lambda_1 = s_{11}\sqrt{\frac{V_{12}}{2V_{11}}}, \qquad \lambda_2 = s_{22}\sqrt{\frac{V_{12}}{2V_{22}}}. \tag{251}$$

A classical identity for jointly Gaussian variables gives

$$\mathbb{E}_Z\big[\Phi(a + bZ)\,\Phi(c + dZ)\big] = \Phi_2\Big(\frac{a}{\sqrt{1+b^2}},\ \frac{c}{\sqrt{1+d^2}};\ \frac{bd}{\sqrt{(1+b^2)(1+d^2)}}\Big). \tag{252}$$

Where $\Phi_2$ is the cdf of the bivariate normal density defined in Appendix (A). Applying this relation to our model yields:

$$\mathcal{Z}_{\text{out}}(y, \omega, V) = \Phi_2\Big(\kappa_1,\ \kappa_2;\ c\Big) \tag{253}$$

with the compact parameters

$$\kappa_1 = s_{11}\,\frac{\sqrt{2}\omega_{11} - \omega_{12}}{\sqrt{2V_{11} + V_{12}}}, \qquad \kappa_2 = s_{22}\,\frac{\sqrt{2}\omega_{22} - \omega_{12}}{\sqrt{2V_{22} + V_{12}}}, \tag{254}$$

$$c = s_{11}s_{22}\,\frac{V_{12}}{\sqrt{(2V_{11} + V_{12})(2V_{22} + V_{12})}} = s_{11}s_{22}\,\frac{1}{3} \quad (V_{11} = V_{22} = V_{12}). \tag{255}$$

We can hence compute denoising function:

$$(g_{\text{out}}(y, \omega, V))_{ab} = \frac{\partial}{\partial \omega_{ab}} \ln \mathcal{Z}_{\text{out}}(y, \omega, V). \tag{256}$$

Because $V_{ab}$ is $\omega$-independent, the chain rule gives

$$\frac{\partial}{\partial \omega_{11}} \Phi_2(\kappa_1, \kappa_2; \rho_{12}) = \frac{\partial \kappa_1}{\partial \omega_{11}}\,\phi_2(\kappa_1, \kappa_2; \rho_{12}), \quad \frac{\partial \kappa_1}{\partial \omega_{11}} = \frac{\sqrt{2}s_{11}}{\sqrt{2V_{11} + V_{12}}}. \tag{257}$$

The four independent derivatives are therefore

$$g_{\text{out}}(y, \omega, V)_{11} = \frac{\sqrt{2}s_{11}}{\sqrt{2V_{11} + V_{12}}}\,\frac{\phi_2(\kappa_1, \kappa_2; \rho_{12})}{\Phi_2(\kappa_1, \kappa_2; \rho_{12})} \tag{258}$$

$$g_{\text{out}}(y, \omega, V)_{22} = \frac{\sqrt{2}s_{22}}{\sqrt{2V_{22} + V_{12}}}\,\frac{\phi_2(\kappa_2, \kappa_1; \rho_{12})}{\Phi_2(\kappa_1, \kappa_2; \rho_{12})} \tag{259}$$

$$g_{\text{out}}(y, \omega, V)_{12} = -\Big(\frac{s_{11}}{\sqrt{2V_{11} + V_{12}}} + \frac{s_{22}}{\sqrt{2V_{22} + V_{12}}}\Big)\,\frac{\phi_2(\kappa_1, \kappa_2; \rho_{12})}{\Phi_2(\kappa_1, \kappa_2; \rho_{12})} \tag{260}$$

This expression can be compactly rewritten as:

$$g_{\text{out}}(y, \omega, V)_{ab} = \frac{1}{\sqrt{6(Q - q)}}\,\frac{\phi(k_1, k_2, c)}{\Phi(k_1, k_2, c)}\begin{pmatrix} \sqrt{2}s_1 & -(s_1 + s_2) \\ -(s_1 + s_2) & \sqrt{2}s_2 \end{pmatrix}_{ab}, \tag{261}$$

where $\phi(k_1, k_2, c)$ is the p.d.f. of a bi-variate Gaussian with zero mean, variances $1/(1 - c^2)$ and covariance $c/(1 - c^2)$, and $\Phi(k_1, k_2, c)$ is its c.d.f (see Appendix A). Moreover, $s_a = 2y_{aa} - 1$, $k_a = s_a(\sqrt{2}\omega_{aa} - \omega_{12})/\sqrt{6(Q - q)}$, $c = s_1 s_2/3$ and $\omega_{ab} = \sqrt{2q}\,\eta_{ab}$. This is precisely the result shown in the main text in Eq. (29).

We finally compute the state equation corresponding to the output channel, namely:

$$\hat{q} = 4\alpha \mathbb{E}_{\eta, y} \sum_{a \leq b} g_{\text{out}}(y, \omega, V)_{ab}^2 \tag{262}$$

where $\eta_{ab} \sim \mathcal{N}(0, 1)$ for $a \leq b = 1, \ldots, T$ and $y \sim \mathcal{Z}_{\text{out}}(y, \omega, V)$.

### D.7 Generalization error and sequence-to-sequence version of the model

In this section we draw some consideration on the generalization error presented in Eq. (10), in the setting of a self-attention layer as in (3) and its sequence-to-sequence version as in (54).

In the main text, we showed the expression of the Bayes-optimal estimation error. In the case of one layer of self-attention this reads:

$$E_{est} = \frac{1}{d}\|S^* - \hat{S}\|_F^2 = Q - q \tag{263}$$

Regarding the generalization error, we instead aim to compute and plot the different quantity shown in Eq. (10), namely:

$$\mathcal{E}_{\mathrm{gen}}(\hat{y}) = \mathbb{E}_{\mathcal{D},S^*}\mathbb{E}_{y_{\mathrm{new}},\boldsymbol{x}_{\mathrm{new}}}\|\hat{y}(\boldsymbol{x}_{\mathrm{new}},\mathcal{D}) - y_{\mathrm{new}}\|_F^2 , \tag{264}$$

with:

$$\hat{y}_{\mathcal{D}}^{\mathrm{BO}}(\mathbf{x}_{\mathrm{test}}) := \mathbb{E}\left[y_{\mathrm{test}} \mid \mathbf{x}_{\mathrm{test}},\mathcal{D}\right] = \int \mathbb{E}_{\mathbf{z}}\left[f_{\mathbf{S}}(\mathbf{x}_{\mathrm{test}})\right]\mathbb{P}(\mathbf{S}\mid\mathcal{D})\mathrm{d}\mathbf{S}$$

Recalling the fact that, for one layer of self-attention, we simply have the relation $y = \sigma_\beta(h) = \sigma_\beta(\{h_{ab}/\sqrt{2-\delta_{ab}}\}_{ab})$, we can introduce the change of variables $h_{ab} = \frac{x_a^\top S x_b - \delta_{ab}\operatorname{Tr}S}{\sqrt{d}}$ and get the expression:

$$\mathcal{E}_{\mathrm{gen}} = \sum_{a,b}\mathbb{E}_{x_{ab}}\int dh_{ab}d\hat{h}_{ab}\|\sigma(\frac{h_{ab}}{\sqrt{2-\delta_{ab}}}) - \sigma(\frac{\hat{h}_{ab}}{\sqrt{2-\delta_{ab}}})\|^2\delta(h_{ab} - \frac{x_a^\top S x_b - \delta_{ab}\operatorname{Tr}S}{\sqrt{d}})\delta(\hat{h}_{ab} - \frac{x_a^\top \hat{S} x_b - \delta_{ab}\operatorname{Tr}\hat{S}}{\sqrt{d}}) \tag{265}$$

We now exploit the fact that, as we know, the preactivations concentrate to:

$$\mathbb{E}_{x_{ab}}\delta(h_{ab} - \frac{x_a^\top S x_b - \delta_{ab}\operatorname{Tr}S}{\sqrt{d}})\delta(\hat{h}_{ab} - \frac{x_a^\top \hat{S} x_b - \delta_{ab}\operatorname{Tr}\hat{S}}{\sqrt{d}}) = \mathcal{N}\left(\begin{pmatrix}h_{ab}\\\hat{h}_{ab}\end{pmatrix},\begin{pmatrix}0\\0\end{pmatrix},\begin{pmatrix}q & q\\q & Q^*\end{pmatrix}2\right) = P(h_{ab},\hat{h}_{ab}) \tag{266}$$

Then, the overall generalization error is given by

$$\mathcal{E}_{\mathrm{gen}} = \sum_{a,b=1}^{T}\mathbb{E}_{(h_{ab},\hat{h}_{ab})\sim P(h_{ab},\hat{h}_{ab})}\left[\sigma(\{\frac{h_{ab}}{\sqrt{2-\delta_{ab}}}\}_{ab}) - \sigma(\{\frac{\hat{h}_{ab}}{\sqrt{2-\delta_{ab}}}\}_{ab})\right]^2 \tag{267}$$

which exactly matches the result presented in the main text in Eq. (19), where the extension to the multi-layer setting is trivial.

Now we slightly modify our model of a self-attention layer by considering its sequence-to-sequence (seq2seq) version $y = \sigma_\beta(\{\frac{h_{ab}}{\sqrt{2-\delta_{ab}}}\}_{ab})x \in \mathbb{R}^{T\times d}$. In particular we aim to compute and plot the generalization error of Eq. (10) in this new setting.

To do so, we define $y = Ax$ and $\hat{y} = \hat{A}x$ with $A = \sigma_\beta(h)$ and $\hat{A} = \sigma_\beta(\hat{h})$ where we leave the factor $\sqrt{2-\delta_{ab}}$ implicit. We exploit the concentration of our input data, in order to compute the Frobenius norm of $y - \hat{y} = (A - \hat{A})x$. Recalling the fact that the input data are iid with $x_{ai}^\mu \sim \mathcal{N}(0,1)$, we use the fact that

$$\sum_{i=1}^{d}x_{t'i}x_{t''i} \approx \delta_{tt'} \tag{268}$$

with high probability when $d$ is large. Hence:

$$\|(A-\hat{A})x\|_F^2 = \sum_{t,i}[\sum_{t'}(A_{tt'} - \hat{A}_{tt'})x_{t'i}]^2 = \sum_{t,i}\sum_{t',t''}(A_{tt'} - \hat{A}_{tt'})(A_{tt''} - \hat{A}_{tt''})x_{t',i}x_{t'',i} \tag{269}$$

but using the concentration property of x we finally get:

$$\|(A-\hat{A})x\|_F^2 = \sum_{t,i}\sum_{t',t''}(A_{tt'} - \hat{A}_{tt'})(A_{tt''} - \hat{A}_{tt''})x_{t',i}x_{t'',i} = \sum_{t,t'}(A_{tt''} - \hat{A}_{tt'})^2 = \|A-\hat{A}\|_F^2 \tag{270}$$

We hence have shown that in the case of $L = 1$ layer, the sequence-to-sequence version of the model shows the same identical state evolution with respect to a single self-attention layer.

### D.8 Details on the numerical implementation

The code used to produce all the figures and the experiments is available at `https://github.com/SPOC-group/ExtensiveRankAttention`. Our gradient descent experiments are done in PyTorch 1.12.1 by minimizing the following loss using Adam

$$\mathcal{L}(W) = \sum_{\mu=1}^{n} \sum_{a,b=1}^{T} \left( y_{ab}^{\mu} - \sigma_\beta \left( \frac{\boldsymbol{x}_a^{\mu\top} W W^\top \boldsymbol{x}_b^{\mu} - \delta_{ab} \operatorname{Tr} W W^\top}{\sqrt{r}\, d} \right) \right)^2 , \tag{271}$$

In our implementation we sample both the input data and the weights of the target as standard Gaussian. Notice that we appropriately adjusted the loss to be consistent with the main test. We choose a learning rate $0.1$ and keep the other hyperparameters at their default parameters and initializing the weights as a standard Gaussian.

When running the averaged version of the algorithm we run the optimization procedure $32$ times for a fixed experiment, and average the matrix $S = W W^\top / \sqrt{rd}$ at the end of training.

Regarding the state equations in the two $L = 1$ cases of softmax and hardmax output channel: in the former case, we simply find the fixed points iterations of the state equations in Eq. (26) and Eq. (233). In the latter case, finally, we compute the expectation in Eq. (262) with Monte-Carlo over $n_{\text{samples}} = 20000$ samples. In particular, to allow for more stable results, we iterate the state equations for $T = 150$ iterations and we compute the mean overlap over the last $30$ iterations of the state equations.

