# OpenReview forum: "Bayes optimal learning of attention-indexed models"
_NeurIPS.cc/2025/Conference — NeurIPS 2025 poster_

### Official Review · Reviewer_VRk2 · 2025-06-23

**Clarity:** 2
**Significance:** 3
**Originality:** 3
**Rating:** 5
**Confidence:** 2

**Summary:**

In this paper, the authors have introduced the Attention-Indexed Model (AIM) to analyze learning in deep attention layers. Theoretically, the authors analyze this using statistical mechanics and random matrix theory to get the closed-form predictions for Bayes-optimal generalization error. Also, the authors analyze the case where embedding dimension $d$ goes to infinity and the sample complexity is $\alpha = n/d^2$.

**Questions:**

Please see weaknesses.

**Ethical Concerns:**

["NO or VERY MINOR ethics concerns only"]

**Final Justification:**

In general, I think this is a good paper, and the authors’ responses are convincing. I have no further questions and have chosen to maintain my original score, as I believe it is already sufficiently high.

**Limitations:**

Yes

**Quality:**

3

**Strengths And Weaknesses:**

The strengths of this paper can be summarized as follows:

First, this paper gives a very thorough theoretical analysis. This is the first tractable analysis of attention mechanisms with extensive-width matrices, whereas previous works only focus on narrow attention layers. As far as I can see, all the theoretical analyses are rigorous and correct.

Second, the authors have also conducted experiments and provided an optimal AMP algorithm. In particular, the result has shown that the practical gradient descent achieves similar performance.

Third, the implementation details and code are included. It makes the experimental results being reproducible.

The weaknesses of this paper are summarized as follows:

First, as the authors have mentioned, they only analyze the single layer case. It would be better if the authors can consider including a more detailed discussion on the generalization to the multi-layer case. For example, what are the potential difficulties one may encounter to generalizing it to multi-layer case?

Second, the author claims that they analyze the case where $d$ is very large. However, in their experiments, $d = 100$ or $d = 200$ seems pretty small or moderate. Is it possible to try larger $d$’s to make it more aligned with the theoretical results?

---

> ### Author Rebuttal · Authors · 2025-07-30
>
> 1. *Generalization to the multi-layer case*:
>
> We point to the prior-channel equations and the related denoising problem with correlated noise (for $L>1$ layers with $L=O(1)$) in Eq. (20) and the paragraph around it. This is actually an interesting random matrix theory problem that may be tractable, but would constitute a result of its own interest in matrix denoising problems. We will emphasize this interesting technical open problem even more in the revised version.
>
> 2. *Large dimension in numerical experiments*:
>
> As the reviewer correctly pointed out, the analytical results obtained are valid in the limit of large embedding dimension $d\to \infty$. Analogously, we also send to infinite the number of samples $n$ and the width of each weight matrix $r_\ell$, with the particular scalings $\alpha = n/d^2 =O(1)$ and $\rho_\ell = r_\ell /d =O(1)$. Regarding the numerical experiments both for AMP, for gradient descent and its averaged version: we use moderate sizes for the embedding dimension, namely $d=100$, and we obtain curves that are strikingly compatible with the Bayes-Optimal theoretical ones (obtained for $d\to \infty$).
> We point out the fact that reproducing the large d results already at such moderate sizes is exactly one of the main points that go in favor of the high-dimensional theoretical approach.
>
> In the revised version, we will add a figure showing that as $d$ increases the numerical experiments will get closer to the theoretical prediction at $d\to \infty$. We expect to be able for explore the range of $d$ from $30$ up to $1000$ for some of the small values of $\rho\sim0.2$, which is not far from the embedding dimensions used in practice.

---

> > ### Comment · Reviewer_VRk2 · 2025-08-02
> >
> > Thank you very much for the clarifications. I have no further questions and will maintain my original score.

---

### Official Review · Reviewer_MHZz · 2025-07-02

**Clarity:** 3
**Significance:** 4
**Originality:** 3
**Rating:** 5
**Confidence:** 4

**Summary:**

The paper proposes the Attention-Indexed Model, designed to study learning in attention-like architectures, where tokens interact with the model parameters as in the attention mechanism of modern Transformers. For the proposed model, the authors derive the generalisation error in the high-dimensional limit of the number of samples $n$, embedding dimension $d$, and width $r$ (with suitable ratios between these quantities) for the Bayes-optimal estimator. They also propose an AMP algorithm that recovers the overlaps appearing in the asymptotic error, thus matching the Bayes-optimal performance. While the theory covers a general $L$-layer model, the authors also discuss the $L = 1$ case as analytically more tractable.

**Questions:**

The authors propose a mean-centered attention mechanism with $1/\sqrt{d}$ scaling. It would be instructive to discuss the alternative $1/d$ scaling (probably, no mean centering needed in this case). Under this "LLN" scaling, are the asymptotics different, or would the current proofs carry over?

**Ethical Concerns:**

["NO or VERY MINOR ethics concerns only"]

**Final Justification:**

The paper provides a novel treatment of the generalisation error in the high-dimensional limit for attention-like models. Novelty of the results and relevance in the modern theory of deep learning are the main reason I am in favor of acceptance.

**Limitations:**

yes

**Quality:**

4

**Strengths And Weaknesses:**

## Strengths

- Novel modeling perspective. By pulling out the bilinear attention indices and studying them directly, the paper isolates what is mathematically distinctive about attention and carries out standard techniques to study the asymptotic behavior. Thus, I see the main contribution as the novelty and relevance of the results more than in their technique.

- I particularly appreciated the discussion on the intractability of the integrals involved which prevent to derive a closed form for the Result 3.1.

- Concrete formulas in the single-layer case. For $L=1$, they can write the AMP’s prior-channel denoiser in closed form via rotationally-invariant eigenvalue shrinkage (RIE) (22) and an explicit state evolution equation (24), leading to analytic phase-transition locations that perfectly match AMP simulations.

## Weaknesses:

I am overall in favor of clear acceptance, as I find the paper technically sound, the results are relevant and novel. These minor weaknesses are to be seen more as suggestions or discussion points

- Background on AMP and Bayes optimality. I think the authors could have given more background on approximate message passing and more explicitly state its connection to Bayes optimal estimation. The paper relies heavily on this equivalence but only mentions it briefly (Appendix C). A short primer in the main text would make the results accessible to a broader audience.

- The paper writes the Bayes free-entropy (or “potential”) of the Attention-Indexed Model as an extremisation problem. Unfortunately, the extrema for $q$ and $\hat{q}$ in Result 3.1 are in implicit for, and it’s thus a bit hard to interpret. However, the authors extenively discuss why the intrgral in the partition function $I_{in}$ is intractable.  However, the authors provide more “meat” with the discussion of the $L=1$ case, which is tractable because now the $\Delta_\ell$ becomes diagonal. I think this compensate for this lack of interpretability.

---

> ### Author Rebuttal · Authors · 2025-07-30
>
> 1. *Background on AMP and Bayes optimality*:
>
> Thank you for this suggestion. Indeed, this is a key point and we will explain it better in the text. The analogy works on the two following lines. On one side we express the posterior distribution of the model with respect to S, which can be manipulated in the considered limit to end up with an optimization problem of a free-entropy in Eq. (15). On the other side, a specific AMP algorithm provably reaches the BO error. The specificity is in the choice of two denoising functions that are taken in such a way that the AMP fixed points are extremizers of the same free-entropy, namely its iterates follows the state evolution equations in Eq. (19). We will make such connections clearer in a revised version of this work.
>
> 2. *Interpretation of the order parameters*:
>
> Regarding the interpretation of the order parameters $q$ and $\hat{q}$: $q$ plays the role of an overlap and, at the fixed points of the free-entropy, is ${\rm Tr}(\hat{S}S^\star)$. On the other hand, $\hat{q}$ is its conjugate parameter and plays the role of a signal-to-noise ratio of a high-dimensional denoising problem in the prior channel.
>
> The fixed point equations for $q$ and $\hat{q}$ are indeed complex for $L>1$ (discussed around Eq. (20)). For $L=1$ they are simpler and numerically easy to evaluate, and hence we focus the discussion of the results in Sec. 4 on this already rich and interesting case.
>
> 3. *Scaling used*:
>
> We are interested in studying the leading order behavior of the attention index. Given that we take the input data to have mean zero and variance one, we expect the dot-product attention to also be a collection of $T\times T$ random variables with mean zero and variance one. In the Bayes-Optimal setting and for our model, dividing the dot product by $d$ instead of $\sqrt{d}$ is just a matter of conventions, as the statistician knows about it and can map between the two by rescaling the inputs.
>
> If the data has a mean of order one, the $1/d$ scaling makes fluctuations in the dot-product attention subleading and the output would concentrate. This would significantly simplify the problem, and the relevant sample complexity scaling will be much lower than $\mathcal{O}(d^2)$. We thus have not considered this case of a target function.
>
> There may be other combinations of scalings that make the problem non-trivial. We are happy to consider them if the referee is curious about any particular one and let us know.

---

> > ### Comment · Reviewer_MHZz · 2025-08-05
> > **Response**
> >
> > I thank the authors for their rebuttal. I was just curious whether a $1/d$ scaling (that is sometimes proposed in the literature) would make the problem more tractable, and I thank the authors for providing a good intuition as to how to think about it.
> >
> > I am still in favor of acceptance.

---

### Official Review · Reviewer_kAGS · 2025-07-02

**Clarity:** 1
**Significance:** 2
**Originality:** 2
**Rating:** 4
**Confidence:** 1

**Summary:**

The paper studies Bayes-optimal generalisation error in a theoretical model of a Transformer. The advantage of their proposal is that they can loosen the restriction on the dimension of queries and keys being small. The provide synthetic experiments to validate their theoretical work.

**Questions:**

I believe it's a typo, but should it not be $x_a^T S_l x_b$ to give a scalar in Equation 1? It looks like $x_a \in \mathbb{R}^d$ is a column vector and $h_{ab}^{\ell}$ a scalar.

Could the $L$ "attention indices" be interpreted as $L$ individual heads? I am finding the choice of notation quite confusing when compared to the notation used in standard transformer works, for example $L$ is usually the number of layers while $H$ is the number of heads, $h$ could maybe be called $\alpha$ to signify that it is acting like an attention coefficient.  -- of course this is personal preference but I am finding the current notation hard to parse .

Equations 4 and 5 are also difficult to parse. How would they resolve in the context of a ViT for example or an LLM in your notation?

**Ethical Concerns:**

["NO or VERY MINOR ethics concerns only"]

**Final Justification:**

Unfortunately my confidence on this paper is quite low so please consider reviewers with higher confidence.

**Limitations:**

The limitations and overall the wording in the paper is often quite fair and honest.

**Quality:**

3

**Strengths And Weaknesses:**

Strengths
----------------
The paper tackles an interesting problem of studying generalisation in Transformers. The assumptions the authors make are reasonable and seem to push forward the surrounding literature (although I am not an expert in this area).

The experimental results seem quite convincing and I appreciate the numerical validation of the theory.

Weaknesses
----------------
The paper targets a very specific community and is hard to follow. I have some light background in RMT and found this quite challenging to read. I believe it would be useful to contextualize with some concrete examples the notation (e.g. Equations 4 and 5). For instance, how would a ViT or LLM look like using the notation. I understand it would not match 1-to-1 with actual Transformer architectures, but at least it would help clarify the notation.

I believe the claim "AIM offers a solvable playground for understanding learning in modern attention architectures" in the abstract is perhaps over-claiming as I do not believe there is high semblance in the update equations to modern attention architectures (a lot of very non-linear and complicated components are missing).

Overall, having some light background in RMT, I found the paper very hard to follow and I think that the paper could benefit from a pass to try to make it more accessible more broadly. I do understand that the nature of the work might make it challenging, for this reason I am happy for the paper to be accepted if other reviewers believe it is worthwhile as I am not overly familiar with this area of research and cannot confidently assess the contribution of the submission given what is already out there.

---

> ### Author Rebuttal · Authors · 2025-07-30
>
> 1. *Weakness 1. and Question 3. : Actual Transformer, and eqs. (4) and (5).*
>
> We thank the referee for the pointers regarding notation. We introduce an abstract attention-indexed model (AIM) in eqs. (1) and (2).
> Then, in eq. (3) we state a simplified multi-layer self-attention with a single head in every layer and no MLP layers. An actual Transformer architecture, such as used in ViT or LLM, would be very close to eq. (3) with multiple heads and attention layers interlayed with token-wise MLP layers. ViT and LLM are usually trained by next-token prediction while the task we consider is supervised.
> Equations (4) and (5) are then a result detailed in appendix B of how to map eq. (3) on (1-2), i.e. how to map the multi-layer self-attention on the AIM. These equations indeed do not look very transparent, but the existence of such mapping is our key contribution as it enables results from the AIM to apply to multi-layer attention. We will clarify this in the revised version.
>
> Architectures with multiple heads could also be mapped to the AIM form and then the number of indices $L$ in the AIM would be the total number of heads in all the layers.
> The only true limitation of the mapping of the transformer architecture to the AIM are the MLP layers. We did not find an analogue of Eqs (4) and (5) that would map the transformer including the MLP layer to the AIM.
>
> 2. *Weakness 2. : AIM claim*:
>
> We agree that "AIM offers a solvable playground for understanding learning in modern attention architectures" is a misleading choice of wording. We will rather say: "AIM offers a solvable playground for understanding learning in self-attention layers, that are key components of modern architectures"
>
>
> 3. *Question 1. : Row-vectors notation*
>
> The notation we used for the quantity $x_a S_\ell x_b^\top$ is correct, as we consider each token $x_a$ to be a row-vector, namely the rows of the whole sequence matrix $X\in \mathbb{R}^{T\times d}$. We understand however that this is in contrast with standard notation of considering column vectors if not better specified. We will change the transposes and make the notation conform to standard practices of vectors being columns in a revised version of this work.
>
>
> 4. *Question 2. : Interpretation of the attention indices L*:
>
> In this work we focus on multi-layer self-attention (3) where $L$ indicates the number of layers, and thus $\ell$ is a layer-index with $\ell=1,\dots,L$.
> However, a collection of attention heads can also map to the AIM. We will add an appendix with such a case and refer to it from the main text to illustrate the generality of the AIM. The observation of the reviewer is indeed spot-on and contained in our analysis.

---

> > ### Comment · Reviewer_kAGS · 2025-08-01
> >
> > Thank you for your response. I will keep my score, but will keep my confidence of 1 as unfortunately the paper is very much outside of my area of expertise.

---

### Official Review · Reviewer_5j3j · 2025-07-23

**Clarity:** 3
**Significance:** 3
**Originality:** 3
**Rating:** 5
**Confidence:** 3

**Summary:**

This work introduces attention-indexed models (AIMs) as a theoretical framework to study learning in attention-based models. These models are the analog of single- or multi-index models for MLPs, adapted to the attention setting. The authors formalize the problem of learning an $L$-layered attention-indexed model, which covers the case of stacked attention layers with residual connections. They characterize the Bayes-optimal predictor in the high-dimensional limit where embedding dimension $d$, sample size $n$ (prop. to number of parameters $d^2$), and attention index matrix rank $r$ grow proportionally, while the number of layers $L$ and context length $T$ remain finite. An approximate message passing algorithm is proposed that recovers the Bayes-optimal performance under certain conditions. The Bayes-optimal analysis is worked out in full detail for the single-layer case ($L=1$), with results applied to both hardmax and softmax attention regimes.

**Questions:**

Please see the section above.

**Ethical Concerns:**

["NO or VERY MINOR ethics concerns only"]

**Final Justification:**

I believe this is a good work that puts forward a nice analogous model to single/multi-index models for attention, that can be used by future works to study different aspects (optimization, more complex transformer architectures—with MLPs, multi-head attention, etc.). In my initial review my, main questions were about interpreting results in the $T \to \infty$ regime—i)whether authors expect a quantitative change ii) relevance of the finite $T$ regime, iii) intuition on results in the hardmax vs softmax case. The authors improved my understanding on i) and iii), and agreed that ii) is a limitation that future works could look into, which is fine in my opinion, as this one of the first works studying properties of such planted models. As a result, I bumped my score from a 4 to a 5.

**Limitations:**

Yes

**Quality:**

3

**Strengths And Weaknesses:**

**Strengths**: This work introduces a new class of models inspired by multi-index models in MLPs to study attention layers. I like the idea of studying the equivalent here given the prevalence of the transformer. The paper takes good first steps by characterizing the generalization and estimation error of the Bayes-optimal solution in this class of models. The results also seem very interesting, especially the contrast between the hard and softmax regimes. There are several directions for future work building on this, including extending to multi-layer attention, adding other architectural components, studying ERM with GD, and extending to symmetric matrices, many of which the authors have already mentioned in the paper.

**Weaknesses**:  I want to highlight a few missing points and some related questions here.

1. **Finite vs. Infinite Context Length Regimes**: You study the regime where context length $T$ is finite. Given that real-world transformers operate on increasingly long contexts, have you considered the asymptotics of $T \to \infty$? Do you expect qualitatively new phenomena to arise in this regime?

2. **Relevance and Plausibility of Scaling Regime**: In your analysis, $d \to \infty$, $n \propto d^2$ and $r \propto d$, while $T$, $L$ remain fixed. I understand that taking $d \to \infty$ helps to apply tools from random matrix theory, but in practice, $d$ is fixed and moderate, while $nT≫d^2$. How should we interpret the Bayes-optimal results in this context? Do phenomena like phase transitions, sample complexity thresholds, or rank dependencies remain qualitatively valid when these asymptotics don’t strictly hold? Are there empirical observations or architectural patterns in real transformers that support the proportionality assumptions you've adopted? A discussion on this would be helpful.


3. **Hardness of $L>1$ Case**: Lines 193–202 say the difficulty arises from cross-layer noise correlations; if the covariance were diagonal the problem would factorize. Could you clarify exactly which analytical steps break down once this correlation is present, and whether any near-diagonal or orthogonal-layer approximations (e.g., treating weak off-diagonal terms perturbatively) might make an $L ⁣> ⁣1$ extension tractable?

4. **Intuition Behind Hardness of Hardmax**: Fig. 2 shows that softmax reaches zero Bayes-optimal error at finite sample complexity, while hardmax never does. Is the barrier for hardmax purely information-theoretic, i.e. the arg-max mapping discards logit magnitudes so the estimator cannot reconstruct them, or is there an additional effect from the activation’s non-smoothness? A brief explanation of which aspect drives the strictly positive asymptotic error would help clarify the contrast with softmax.


5. **Sharp Transition in Small Width Limit**: In Corollary 4.2 (Fig. 1 right), you show a weak recovery threshold for hardmax with small width. Could you provide intuition for why performance is flat below this threshold, i.e., the estimator behaves like it has no signal? What changes fundamentally at the threshold?

6. **Rank Sensitivity in Hardmax vs. Softmax**: Figure 2 shows that lowering the rank makes the softmax task noticeably easier, its strong-recovery threshold drops. Whereas Figure 1 suggests the hardmax error curve barely shifts with rank. Why does softmax benefit so strongly from reduced rank while hardmax remains almost insensitive?

---

> ### Author Rebuttal · Authors · 2025-07-30
>
> We sincerely thank the reviewers for their questions. We will reflect our answers in the revised manuscript to improve its clarity.
>
> 1. *Finite vs. Infinite Context Length Regimes*:
>
> Our theoretical results indeed require $T<<d$. However, we can take the limit $T\to \infty$ after the high-dimensional limit. In this case, we do not expect a qualitative change. E.g. the location of the phase transition just requires a rescaling of the sample complexity, see Fig. (2) right panel. Indeed, notice that in Result (4.3) we specify Eq. (19) for $\hat{q}$ in the case of a softmax channel, obtaining a simple expression which do not depend on the number of tokens anymore once we do the simple rescaling $\bar{\alpha}=\alpha (T^2 +T-2)$.  A qualitative change would likely arise with more complex structures of the data; proposing a suitable model along that line is an interesting avenue of future exploration.
>
> 2. *Relevance and Plausibility of Scaling Regime*:
>
> In general, one would want to explore as rich range of scalings as possible. Previous work [22-24] treated the case $d \to \infty$, and $r, T, L$ finite. In this respect, our paper explores a more general case that allows $r \propto d$ as often chosen in practice. The choice of the sample complexity is given by the fact that it is in this scaling that the performance changes from trivially bad to perfect. Our requirement of $T, L$ remaining finite is indeed a limitation that future work will hopefully be able to remove.
> One point to make is that while our theory results are derived in the $d \to \infty$ limit, the results do apply for very moderate values of $d$, see Fig. 1 where experiments with AMP are performed with $d=100$. Analogously, GD and its averaged version also show the predicted recovery thresholds (found theoretically for infinite d) with only $d=100$. The condition asked by the reviewer $nT>>d^2$ corresponds as $\bar{\alpha}T>>1$, so technically this scaling is already contained in our analysis by taking $\bar{\alpha}$ very large.  We will clarify the discussion about the reasoning behind the considered scaling limit in the revised paper.
>
> 3. *Hardness of $L>1$ Case*:
>
> For $L>1$ the off-diagonal terms of the overlap matrix are of the form ${\rm Tr}(S_1 S_2)$, which do not have mean zero but scale with $\rho^2$ and is thus not small for moderate $\rho$. Diagonal uncoupling could be present in the low $\rho$ regime, where a perturbative approach could be possible, but we were mainly interested in varying this width-ratio parameter and investigate its impact on the properties of the model. Extending to the $L>1$ case would imply solving a high-dimensional denoising problem with correlated noise for the prior channel, which we presently do not know how to do. We left this for future work and progress in random matrix theory.
>
> 4. *Intuition Behind Hardness of Hardmax*:
>
> Generating labels using a hardmax output channel corresponds to learning binary associations over the token sequence $X$, with discrete labels $y \in \{0,1\}$. This task is intrinsically harder than the softmax case. Conceptually, this mirrors the difference between binary classification with a single-layer perceptron—where labels are generated by applying a sign function to a random linear projection—and linear regression on a linear target. In the classification case, perfect generalization typically requires more than a number of samples linear in the input dimension. By contrast, in the regression setting, zero test error can be achieved once the number of samples exceeds the input dimension. In this analogy, the softmax behaves like the linear regression case, while the hardmax mimics the more difficult binary classification case. We will reflect this discussion in the revised paper.
>
> 5. *Sharp Transition in Small Width Limit*:
>
> The mechanism behind the weak-recovery threshold is that of the BBP transition. When the width of the model decreases and becomes of order $r=O(1)$ in the dimension d, the problem is related to that of a rank-1 spike denoising. This can be noticed mathematically in the computations in Appendix D.2. The spike is contained in the bulk before some $\bar{\alpha}_c$ value in this small width regime. The spike instead emerges from the bulk above such value, namely the MMSE starts decreasing and some information starts being recovered even at finite width $r=O(1)$.
>
> 6. *Rank Sensitivity in Hardmax vs. Softmax:*:
>
> In case the referee was commenting on Fig. (1) left: both for the hard-max and soft-max the task is easier as lower rank (Fig. 2 left and Fig. 1 left). We point out that the scale of Fig. (2) is logarithmic and that the difference in MMSE among the various curves is more pronounced when visualizing the curves in linear scale as in Fig. (1).  In case the referee was commenting on Fig. (1) right: Note the x-axes in Fig. (1) right is rescaled by r to illustrate better the low-r limit. But Fig. (1) right cannot be interpreted as the task getting harder as the rank is lowered.

---

> > ### Comment · Reviewer_5j3j · 2025-08-08
> >
> > Thanks to the authors for their patience, sorry for being late with my response. Thanks for answering my questions (which were mainly about $T \to \infty$ regime), and also improving my intuition on hardmax vs softmax. I think the discussions (on the plausibility of scaling regime and hardmax vs softmax intuition, and the nice connection to classification and regression that the authors made) would help the paper (so please include them in the revised version). I have also read other reviews and author responses, and I am happy to bump my score to a 5.

---

> ### Comment · Area_Chair_zopk · 2025-08-06
> **Please Read Authors' Rebuttal and Respond ASAP**
>
> There are only two days remaining in the author-reviewer discussion phase. The authors have made a concerted effort to address your concerns, and it's important that you respond to let them know whether their rebuttal addresses the issues that you raised.

---

### Decision · Program_Chairs · 2025-09-17

**Decision:**

Accept (poster)

**Comment:**

This paper proposes the Attention-Indexed Model (AIM), a theoretical framework for studying learning in deep attention layers. AIM generalizes prior tractable models by allowing full-rank key and query matrices, making it more faithful to modern Transformer architectures. The paper derives closed-form predictions of Bayes-optimal generalization error and characterizes phase transitions in learning as functions of embedding dimension, sample complexity, width, and sequence length. The paper further proposes an Approximate Message Passing (AMP) algorithm that achieves Bayes-optimal performance under certain conditions, and demonstrates that gradient descent can converge to achieve optimal performance. Synthetic experiments validate the theoretical predictions.

The paper was reviewed by four referees who agree on the paper's primary strengths and weaknesses. The AC and reviewers find that the proposed AIM framework provides a novel, technically sound means of analyzing generalization error and phase transitions in the high-dimensional limit for attention-based models. All four reviewers emphasize the significance of the results, which are very thorough, interesting, and convincing. The framework and the theoretical results contribute to our collective understanding of learning in architectures with deep attention layers and provide several interesting directions for future work. Some reviewers raised concerns with the original submission, noting that the presentation would benefit from improved accessibility. In particular, they suggested drawing explicit connections to architectures such as LLMs and ViTs to better contextualize the work, and including background material on Bayes optimality and AMP in the main paper. The authors addressed the reviewers' primary questions and concerns during the rebuttal period, as reflected in the reviewers' Final Justification.